# Folding of prestin's anion-binding site and the mechanism of outer hair cell electromotility

Xiaoxuan Lin[1,2], Patrick R Haller[1,2], Navid Bavi[1,2], Nabil Faruk[2], Eduardo Perozo[1,2,3,4]*, Tobin R Sosnick[1,2,4,5]*

[1]Center for Mechanical Excitability, The University of Chicago, Chicago, United States; [2]Department of Biochemistry and Molecular Biology, The University of Chicago, Chicago, United States; [3]Institute for Neuroscience, The University of Chicago, Chicago, United States; [4]Institute for Biophysical Dynamics, The University of Chicago, Chicago, United States; [5]Prizker School for Molecular Engineering, The University of Chicago, Chicago, United States

*For correspondence:
eperozo@uchicago.edu (EP);
trsosnic@uchicago.edu (TRS)

Competing interest: The authors declare that no competing interests exist.

**Abstract** Prestin responds to transmembrane voltage fluctuations by changing its cross-sectional area, a process underlying the electromotility of outer hair cells and cochlear amplification. Prestin belongs to the SLC26 family of anion transporters yet is the only member capable of displaying electromotility. Prestin's voltage-dependent conformational changes are driven by the putative displacement of residue R399 and a set of sparse charged residues within the transmembrane domain, following the binding of a $Cl^-$ anion at a conserved binding site formed by the amino termini of the TM3 and TM10 helices. However, a major conundrum arises as to how an anion that binds in proximity to a positive charge (R399), can promote the voltage sensitivity of prestin. Using hydrogen–deuterium exchange mass spectrometry, we find that prestin displays an unstable anion-binding site, where folding of the amino termini of TM3 and TM10 is coupled to $Cl^-$ binding. This event shortens the TM3–TM10 electrostatic gap, thereby connecting the two helices, resulting in reduced cross-sectional area. These folding events upon anion binding are absent in SLC26A9, a non-electromotile transporter closely related to prestin. Dynamics of prestin embedded in a lipid bilayer closely match that in detergent micelle, except for a destabilized lipid-facing helix TM6 that is critical to prestin's mechanical expansion. We observe helix fraying at prestin's anion-binding site but cooperative unfolding of multiple lipid-facing helices, features that may promote prestin's fast electromechanical rearrangements. These results highlight a novel role of the folding equilibrium of the anion-binding site, and help define prestin's unique voltage-sensing mechanism and electromotility.

## eLife assessment

This study presents **important** findings regarding the local dynamics at the anion binding site in the SLC26 transporter prestin that is responsible for electromotility in outer hair cells. The authors reveal critical differences to homologous proteins and thereby provide insight into prestin's unique function. The evidence is generally **convincing**, although the interpretations concerning the mechanistic basis for voltage sensitivity would benefit from orthogonal evidence.

## Introduction

Hearing sensitivity in mammals is sharply tuned by a cochlear amplifier associated with electromotile length changes in outer hair cells (*Dallos et al., 2006*). These changes are driven by prestin (SLC26A5),

a member of the SLC26 anion transporter family, which converts voltage-dependent conformational transitions into cross-sectional area changes, affecting its footprint in the lipid bilayer (*Sfondouris et al., 2008*). This process plays a major role in mammalian cochlear amplification and frequency selectivity, with prestin knockout producing a 40- to 60-dB signal loss in live cochleae (*Liberman et al., 2002*). Unlike most molecular motors, where force is exerted from chemical energy transduction, prestin behaves as a putative piezoelectric device, where mechanical and electrical transduction are coupled (*Dong et al., 2002*). As a result, prestin functions as a direct voltage-to-force transducer. Prestin's piezoelectric properties are unique among members of the SLC26 family, where most function as anion transporters.

Recent structures determined by cryo-electron microscopy (cryo-EM) have sampled prestin's conformational space under various anionic environments and located the anion-binding site at the electrostatic gap between the amino termini of TM3 and TM10 helices (*Bavi et al., 2021*; *Ge et al., 2021*; *Butan et al., 2022*). This anion-binding pocket is highly conserved, and is influenced by surrounding hydrophobic residues in TM1 and by a fixed positive charge from residue R399 on TM10. Movements of this binding site are coupled to the complex reorientation of the core domain relative to the gate domainn (*Bavi et al., 2021*; *Ge et al., 2021*), reminiscent of the conformational transitions in transporters displaying an elevator-like mechanism (*Garaeva and Slotboom, 2020*). Prestin exhibits minimal transporter ability yet is structurally similar to the non-electromotive anion transporter SLC26A9 (sequence identity = 34%; Cα root-mean-square deviation (RMSD) = 3.4 Å for the transmembrane domain [TMD], PDB: 7S8X and 6RTC) (*Butan et al., 2022*; *Walter et al., 2019*; *Chi et al., 2020*). Questions remain as to the molecular basis underlying the distinct functions of the two proteins. Importantly, the role of bound anions, which is required for prestin electromotility (*Oliver et al., 2001*), is still elusive.

Prestin's voltage dependence is tightly regulated by intracellular anions of varying valence and structure (*Rybalchenko and Santos-Sacchi, 2003*; *Rybalchenko and Santos-Sacchi, 2008*), whereas anion affinity is also regulated by voltage and tension (*Song and Santos-Sacchi, 2010*). These phenomena suggest that anions, rather than behaving as explicit gating charges, may serve as allosteric modulators (*Song and Santos-Sacchi, 2010*). Incorporating a fixed charge alternative to a bound anion through an S398E mutation preserves prestin's nonlinear capacitance (NLC) but results in insensitivity to salicylate, a strong competing anionic binder (*Butan et al., 2022*). Except for residue R399, charged residues located in the TMD distribute toward the membrane–water interface (*Bavi et al., 2021*; *Ge et al., 2021*; *Butan et al., 2022*) and display minimal contributions to the total gating charge estimated from NLCs (*Bai et al., 2009*). Electrostatic calculations show that R399 has a strong contribution to the local electrostatics at the anion-binding site, by providing ~40% of the positive charge at the bilayer mid-plane (*Bavi et al., 2021*). However, the existing structural and functional data cannot explain why prestin's voltage dependence requires close proximity of both a negative charge (the bound anion or S398E) (*Oliver et al., 2001*) and a positive charge (R399; *Bavi et al., 2021*; *Gorbunov et al., 2014*). The resolution of this conundrum will define an essential step toward our understanding of prestin's unique voltage-sensing mechanism.

Here, we studied the influence of anion-binding on the dynamics and structural changes of prestin as a function of anions ($Cl^-$, $SO_4^{2-}$, salicylate, and HEPES (4-(2-hydroxyethyl)-1-piperazineethanesul fonic acid)) via hydrogen–deuterium exchange mass spectrometry (HDX-MS). The HEPES condition was achieved by $Cl^-$ removal (dialysis), which inhibits prestin's NLC (*Oliver et al., 2001*). In a HEPES-based buffer, prestin's NLC shifts to depolarized potentials, associated to a more expanded state at 0 mV that is coupled to low anion affinity (*Rybalchenko and Santos-Sacchi, 2003*; *Rybalchenko and Santos-Sacchi, 2008*). Based on the above studies and the large size of HEPES anions, we assumed minimal binding of HEPES anions to prestin and hence associated HEPES condition to a putative apo state in this study. By comparing the dynamics of prestin with its close non-piezoelectric relative, the anion transporter SLC26A9, we identified distinct features unique to prestin, including a relatively unstable anion-binding site that folds upon binding, thereby allosterically modulating the dynamics of the TMD. In contrast, the stability and hydrogen-bond pattern of SLC26A9's anion-binding site were minimally affected by anion binding, albeit displaying high similarities to prestin in both structure and sequence. Prestin reconstituted in nanodisc exhibited indistinguishable dynamics compared to detergent-solubilized prestin, except for a destabilized TM6 which mediates prestin's mechanical expansion (*Bavi et al., 2021*). We observed fraying of the helices involved in the binding site whereas

**Table 1.** Biochemical and statistical details for HDX.

| Dataset | Prestin, Cl⁻ | Prestin, HEPES (apo) | Prestin, SO₄²⁻ | Prestin, salicylate | Prestin, nanodisc, Cl⁻ | Slc26a9, Cl⁻ | Slc26a9, HEPES (apo) |
|---|---|---|---|---|---|---|---|
| HDX reaction details | 360 mM NaCl, 20 mM Tris (NaPi for $pD_{read}$ 6.1, 0°C), 3 mM DTT, 1 mM EDTA, 0.02% GDN. $pD_{read}$ 7.1, 25°C or $pD_{read}$ 6.1, 0°C | 150 mM HEPES, 0.02% GDN. $pD_{read}$ 7.1, 25°C | 140 mM $Na_2SO_4$, 5 mM $MgSO_4$, 20 mM NaPi, 0.02% GDN. $pD_{read}$ 7.1, 25°C or $pD_{read}$ 6.1, 0°C | 140 mM $Na_2SO_4$, 5 mM $MgSO_4$, 50 mM salicylate, 20 mM NaPi, 0.02% GDN. $pD_{read}$ 7.1, 25°C or $pD_{read}$ 6.1, 0°C | 20 mM Tris (NaPi for $pD_{read}$ 6.1, 0°C), 150 mM NaCl. $pD_{read}$ 7.1, 25°C or $pD_{read}$ 6.1, 0°C | 360 mM NaCl, 20 mM Tris (NaPi for pD 6.5, 0°C), 3 mM DTT, 1 mM EDTA, 0.02% GDN. $pD_{read}$ 7.1, 25°C or $pD_{read}$ 6.1, 0°C | 150 mM HEPES, 0.02% GDN. $pD_{read}$ 7.1, 25 °C |
| HDX time course (*: replicated. Times in parenthesis: times in $pD_{read}$ 7.1, 25°C after correcting for the $k_{chem}$ difference) | $pD_{read}$ 6.1, 0°C: 1 s (0.007 s), 10 s (0.07 s)*, 90 s (0.6 s)*; $pD_{read}$ 7.1, 25°C: 6 s, 10 s*, 30 s, 90 s*, 5 min, 15 min*, 45 min, 150 min*, 27 hr* | $pD_{read}$ 7.1, 25°C: 10 s*, 90 s*, 5 min*, 15 min*, 150 min*, 27 hr* | $pD_{read}$ 6.1, 0°C: 1 s (0.007 s), 10 s (0.07 s), 90 s (0.6 s); $pD_{read}$ 7.1, 25°C: 10 s, 30 s, 90 s, 5 min, 15 min, 150 min, 27 hr | $pD_{read}$ 6.1, 0°C: 1 s (0.007 s), 90 s (0.6 s); $pD_{read}$ 7.1, 25°C: 10 s, 90 s, 5 min, 15 min, 150 min, 27 hr | $pD_{read}$ 6.1, 0°C: 90 s (0.6 s); $pD_{read}$ 7.1, 25°C: 10 s, 90 s, 15 min, 150 min, 27 hr | $pD_{read}$ 6.1, 0°C: 1 s (0.007 s), 10 s (0.07 s)*, 90 s (0.6 s)*; $pD_{read}$ 7.1, 25°C: 10 s*, 30 s, 90 s*, 5 min, 15 min*, 150 min*, 27 hr* | $pD_{read}$ 7.1, 25 °C: 10 s*, 90 s*, 15 min*, 150 min*, 27 h* |
| HDX control samples | Non-deuterated control; in-exchange control; maximally labeled control | | | | | Non-deuterated control; in-exchange control; maximally labeled control | |
| In- and back-exchange, mean/IQR | In-exchange: 3.1%/2.0%; back-exchange: 27%/14% | | | | | In-exchange: 2.5%/1.9%; back-exchange: 29%/17% | |
| No. of peptides | 266 (TMD: 95; cytosolic: 171) | 265 (TMD: 94; cytosolic: 171) | 266 (TMD: 95; cytosolic: 171) | 265 (TMD: 94; cytosolic: 171) | 256 (TMD: 85; cytosolic: 171) | 338 (TMD: 85; cytosolic: 253) | 335 (TMD: 82; cytosolic: 253) |
| Sequence coverage | 83% (TMD: 75%; cytosolic: 95%) | 83% (TMD: 75%; cytosolic: 95%) | 83% (TMD: 75%; cytosolic: 95%) | 83% (TMD: 74%; cytosolic: 95%) | 79% (TMD: 67%; cytosolic: 95%) | 81% (TMD: 68%; cytosolic: 96%) | 81% (TMD: 68%; cytosolic: 96%) |
| Average peptide length/ redundancy | 12.2/4.3 | 12.2/4.3 | 12.2/4.3 | 12.2/4.3 | 12.2/4.1 | 12.5/5.3 | 12.5/5.3 |
| Replicates | 2 (Biological) | 3 (Biological) | 1 | 1 | 1 | 3 (Technical) | 3 (Technical) |
| Repeatability (TM peptides only) | 0.69%/0.06 Da (average SD of the Δ%D/Δ#D between the duplicates) | 0.93%/0.09 Da (average SD) | N/A | N/A | N/A | 0.64%/0.06 Da (average SD) | 0.60%/0.05 Da (average SD) |
| Significant differences in HDX (ΔHDX > X Da, TM peptides only) | N/A | 0.22 Da (95% CI) | N/A | N/A | N/A | 0.14 Da (95% CI) | 0.13 Da (95% CI) |
| | 0.17 Da (95% CI) | | N/A | N/A | N/A | 0.11 Da (95% CI) | |

cooperative unfolding of multiple lipid-facing helices including TM6, which may explain prestin's fast and large-scale motions. These results highlight the significance of the anion-binding site's folding equilibrium in defining the unique properties of prestin's voltage dependence and electromotility.

## Results

We carried out HDX measurements on dolphin prestin and mouse SLC26A9 solubilized in glyco-diosgenin (GDN) at either $pD_{read}$ 7.1, 25°C or $pD_{read}$ 6.1, 0°C (**Table 1**). The observed HDX rates reported on the stability, as exchange occurred mostly via EX2 kinetics (Appendix 1). Employment of the two conditions increased the effective dynamic range of the HDX measurement to span seven log units, allowing us to determine the stability of both the highly and minimally stable regions within the protein (**Hamuro, 2021**). To properly combine the two datasets, the stability of the protein should be the same under the two conditions, and this was supported by the exchange rates scaling with $k_{chem}$ (intrinsic exchange rates) (**Bai et al., 1993**; **Nguyen et al., 2018**; **Appendix 1—figure 1**).

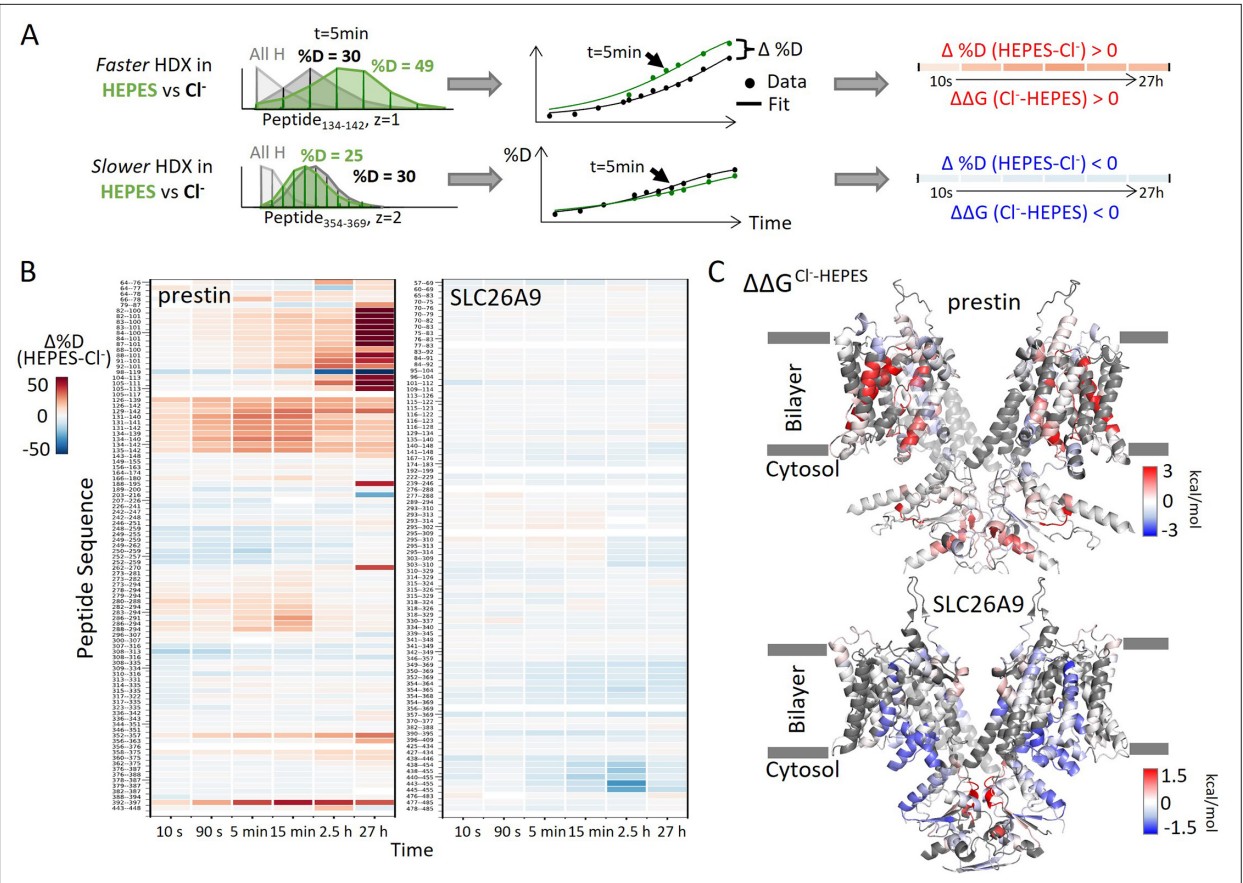

**Figure 1.** Distinct HDX response of prestin and SLC26A9 to Cl⁻ binding. (**A**) HDX data analysis to obtain (**B**) and (**C**). One example peptide is shown in cases where HDX becomes faster or slower in HEPES (the putative apo state) compared to in Cl⁻. Deuteration levels are obtained from the mass spectra. Here, spectra for the undeuterated peptide (gray) and after 5 min HDX labeling in Cl⁻ (black) and HEPES (green) are shown as an example. The resulting deuterium uptake plots are used to generate the differential deuteration heatmaps in (**B**). Changes in free energy of unfolding (ΔΔG) in (**C**) are calculated after fitting the data with a stretched exponential (Materials and methods) (*Hamuro, 2021*). (**B**) Heatmaps showing the difference in deuteration levels at each labeling time for all transmembrane domain (TMD) peptides of prestin and SLC26A9 measured in HEPES compared to Cl⁻. Peptide sequences are displayed on the *y*-scale and legible through the high-resolution image. (**C**) The ΔΔGs in HEPES compared to Cl⁻ for full-length prestin and SLC26A9 mapped onto the structure (PDB 7S8X and 6RTC). Red and blue indicate increased and decreased stability upon Cl⁻ binding, respectively. Following regions of the left subunits are shown as low transparency to highlight the binding site – prestin: TM5 and TM12–14; SLC26A9: TM5 and TM13–14. Regions with no fitting results are in gray.

The online version of this article includes the following figure supplement(s) for figure 1:

**Figure supplement 1.** Volcano plot analysis of HDX for prestin (**A**) and SLC26A9 (**B**) in response to Cl⁻ binding.

**Figure supplement 2.** Deuterium uptake curves for all peptides covering prestin's transmembrane domain.

**Figure supplement 3.** Deuterium uptake curves for all peptides covering SLC26A9's transmembrane domain.

The HDX data were presented in terms of the relevant region with the specific sequence and peptides noted in parentheses (Materials and methods), for example, the N-terminus of prestin's TM10 (Region₃₉₄₋₃₉₇: Peptide₃₉₂₋₃₉₇). Although we mostly focused on the anion-binding site, we also obtained comparative thermodynamic information throughout the two proteins (Appendix 2).

## Prestin's anion-binding site is less stable than SLC26A9s

To examine the effect of anion binding to the dynamics of prestin and SLC26A9, we dialyzed the proteins purified in Cl⁻ into a HEPES buffer lacking other anions. Cl⁻ removal resulted in distinct stability changes for prestin and SLC26A9, manifested by significant HDX acceleration for prestin while mild HDX slowing for SLC26A9 (*Figure 1*, *Figure 1—figure supplement 1*). These HDX effects indicate that anion binding induced global stabilization for prestin while slight destabilization for SLC26A9 (*Figure 1*).

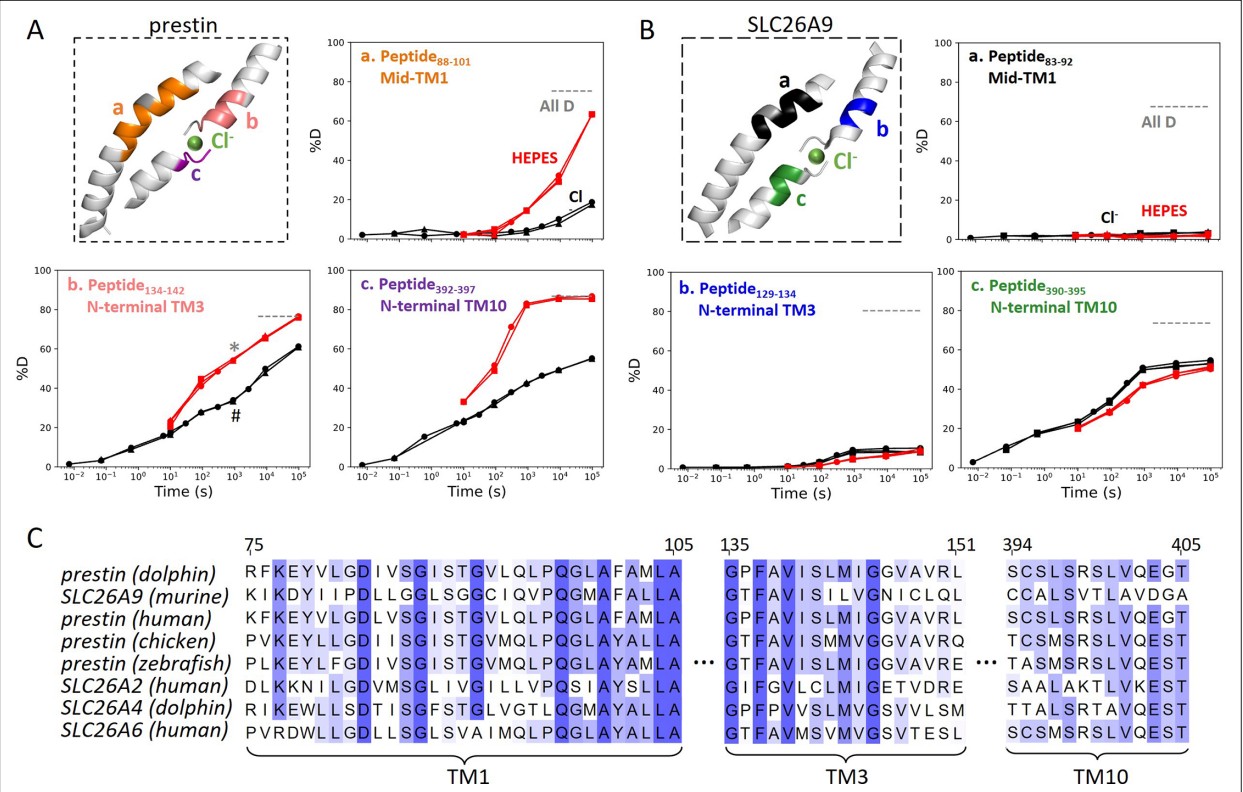

**Figure 2.** The anion-binding pockets for prestin and SLC26A9 exhibit distinct stability changes upon Cl⁻ binding, albeit highly conserved. Cl⁻ binding stabilizes prestin's anion-binding pocket (**A**) but mildly affects SLC26A9s (**B**). The structure shows the anion-binding pocket (TM1, TM3, and TM10) with the putative position of the bound Cl⁻. Colored regions correspond to peptides whose deuterium uptake plots are shown when the protein is in Cl⁻ (black) and in HEPES (red). Prolines are colored in gray. Gray dashed lines indicate deuteration levels in the full-D control. Data from two and three biological replicates are shown for prestin in Cl⁻ and HEPES, respectively. Data from three technical replicates are shown for SLC26A9. Replicates are shown as circles, triangles, and squares. Some replicates are superimposable and hence not observable. The symbols (* and #) in (A.b) denote data points used in *Figure 3B*. (**C**) Sequence alignment using Clustal Omega of prestin and close SLC26 transporters across species for the anion-binding pocket. Shades of blue indicate degree of conservation.

The online version of this article includes the following figure supplement(s) for figure 2:

**Figure supplement 1.** Site-resolved protection factors for prestin and SLC26A9 obtained using PyHDX.

**Figure supplement 2.** Mammalian prestin has a conserved and helix-destabilizing proline 136 on TM3.

Among the observed HDX responses for prestin, the HDX acceleration at the anion-binding pocket appeared to be the most pronounced and indicates local stabilization induced by anion binding (*Figure 2A*). In detail, HDX accelerated by 20-fold for the N-termini of both TM3 (Region₁₃₆₋₁₄₂: Peptide₁₃₄₋₁₄₂ + 9 other peptides) and TM10 (Region₃₉₄₋₃₉₇: Peptide₃₉₂₋₃₉₇) (*Figure 2A*, *Figure 1—figure supplement 2.22–2.31*). This HDX change translates to a difference in free energy of unfolding ($\Delta\Delta G$) by at least 1.8 kcal/mol; $\Delta\Delta G^{\text{bindingsite}}_{\text{Cl binding}} = 1.8$ kcal/mol. At least four residues in the middle of TM1 exhibited faster HDX (Region₉₀₋₁₀₁: Peptide₈₈₋₁₀₁ + 10 other peptides), collectively by 350-fold; $\Delta\Delta G^{\text{TM1}}_{\text{Cl binding}} = 3.5$ kcal/mol (*Figure 1A*, *Figure 1—figure supplement 2.6–2.16*). The TM1 region with accelerated HDX included L93, Q97, and F101, residues that are known to participate in the binding pocket (*Bavi et al., 2021*; *Ge et al., 2021*; *Schaechinger et al., 2011*).

SLC26A9 exhibited similar stability as prestin in Cl⁻ for the majority of the TMD, except for the N-terminal TM3 (Region₁₃₁₋₁₃₄: Peptide₁₂₉₋₁₃₄) which exchanged at least 100-fold slower than that of prestin's (Region₁₃₆₋₁₄₂: Peptide₁₃₄₋₁₄₂) (*Figure 2*). This difference in HDX points to a relatively unstable anion-binding site of prestin as compared to SLC26A9; $\Delta\Delta G^{\text{N−terminalTM3}}_{\text{SLC26A9−prestin}} = 2.8$ kcal/mol, and was also seen in the site-resolved protection factors (PFs) that were obtained by deconvoluting the HDX-MS data using PyHDX (*Smit et al., 2021*; *Figure 2—figure supplement 1*).

Compared to the 20- to 350-fold HDX acceleration observed at prestin's binding site upon $Cl^-$ removal, HDX of SLC26A9's binding pocket was only affected mildly (*Figure 2B*). These included a slight slowing in HDX for the N-termini of TM3 (Region$_{131-134}$: Peptide$_{129-134}$) and TM10 (Region$_{392-395}$: Peptide$_{390-395}$) (*Figure 2B*). The TM1 (Region$_{72-92}$: Peptide$_{83-92}$ + 10 other peptides) continued to remain undeuterated even after 27 hr (*Figure 2*, *Figure 1—figure supplement 3.4–3.14*), emphasizing the intrinsic high stability of SLC26A9's anion-binding pocket.

Although the anion-binding pocket is highly conserved and structurally similar across members of the SLC26 family and SLC26A5 families (*Figure 2C*), mammalian prestin is the only member capable of displaying eletromotility (*Santos-Sacchi and Navaratnam, 2022*). Hence, the distinct stability responses we observe for dolphin prestin and mouse SLC26A9 point to a prestin's unique adaptation as a motor protein.

In addition to the binding pocket, we observed stability changes in various regions of the TMDs for prestin and SLC26A9 that may explain their distinct functions. For prestin, anion binding resulted in stabilization for the intracellular cavity but destabilization for regions facing the extracellular milieu (*Figure 1C*, *Figure 1—figure supplement 1A*). The stabilizing effects for the cytosol-facing regions were manifested by HDX acceleration upon $Cl^-$ removal at the linker between TM2 and TM3, and the intracellular portions of TM7, TM8, and TM9 (Region$_{128-135}$, Region$_{284-294}$, and Region$_{354-375}$) (*Figure 1—figure supplement 2.22–2.27, 2.58–2.64, 2.82–2.87*). In contrast, HDX slowed for the regions facing the extracellular environment, namely the extracellular ends of TM5b, TM6, and TM7 (Region$_{250-262}$ and Region$_{309-316}$) (*Figure 1—figure supplement 2.45–2.51, 2.66–2.68, 2.71*). However, for SLC26A9, anion binding destabilized the cytosol-facing regions, as HDX slowed by ~fivefold upon $Cl^-$ removal for the intracellular ends of TM8, TM9, and TM12 (Region$_{351-369}$ and Region$_{440-455}$) (*Figure 1C*, *Figure 1—figure supplement 1B*, *Figure 1—figure supplement 3.62–3.63, 3.77–3.82*). The distinct thermodynamic consequences of anion binding for prestin and SLC26A9 point to a distinct molecular basis underlying their different functions as a motor and a transporter, respectively.

## Anion binding drives the folding of prestin's binding site

For prestin in HEPES, which adopted the putative apo state, the 20-fold HDX acceleration for the binding site (*Figure 2A*) is consistent with a process of local destabilization, even unfolding, or increased solvent accessibility as the region becomes exposed to the intracellular water cavity (*Bavi et al., 2021*). To investigate these possibilities, we measured prestin's HDX in response to a chaotrope, urea, which destabilizes proteins by interacting with backbone amides (*Lim et al., 2009*). In a background of 360 mM $Cl^-$, the addition of 4 M urea accelerated HDX for the N-terminus of TM3 (Region$_{137-140}$: Peptide$_{134-140}$) by 20-fold (*Figure 3A*), suggesting that this region was destabilized and accessible to urea in its exchange-competent state. In apo prestin, however, the PF at the N-terminus of TM3 was unaffected by urea (after accounting for the known ~50% slowing of the $k_{chem}$; *Lim et al., 2009*; *Figure 3A*), arguing that this region was already unfolded prior to the addition of urea (*Ramesh et al., 2019*).

We note that in apparent contradiction to our inference that the N-terminus of TM3 was unfolded in the apo state, its HDX was ~100-fold slower than $k_{chem}$. Such apparent PF for an unfolded region has been reported when it is located inside an outer membrane β-barrel, rationalized by the region having a lower effective local concentration of the HDX catalyst, $[OD^-]$, than in bulk solvent (*Zmyslowski et al., 2022*; *Lin et al., 2022*). For prestin, we propose that detergent molecules in the micelle can restrict the access of $OD^-$ to amide protons, leading to a local effective pD lower than the bulk solvent and hence producing the apparent PF for the unfolded N-terminus of TM3.

The folding reversibility of the anion-binding site was evaluated by tracking the HDX for 5 min after titrating in $Cl^-$ to apo prestin. Deuteration levels for the N-terminus of TM3 (Region$_{137-140}$) decreased with increasing $Cl^-$ concentration (*Figure 3B*), suggesting reversible folding upon $Cl^-$ binding. Similar behavior was seen in the middle of TM1 (Region$_{86-101}$: Peptide$_{84-101}$) as $Cl^-$ binding stabilized the binding pocket (*Figure 3B*).

We also examined prestin's stability in its intermediate states, obtained by replacing $Cl^-$ anions with $SO_4^{2-}$ and salicylate (*Bavi et al., 2021*; *Ge et al., 2021*). When $SO_4^{2-}$ is the major anion, prestin's HDX was nearly identical as in $Cl^-$, except for a slightly faster HDX at the anion-binding site (Region$_{128-142}$ and Region$_{394-397}$) at labeling times longer than $10^3$ s (*Figure 4*). This mild HDX response suggested a slightly destabilized binding site while the remaining regions retained normal dynamics as in $Cl^-$. In the

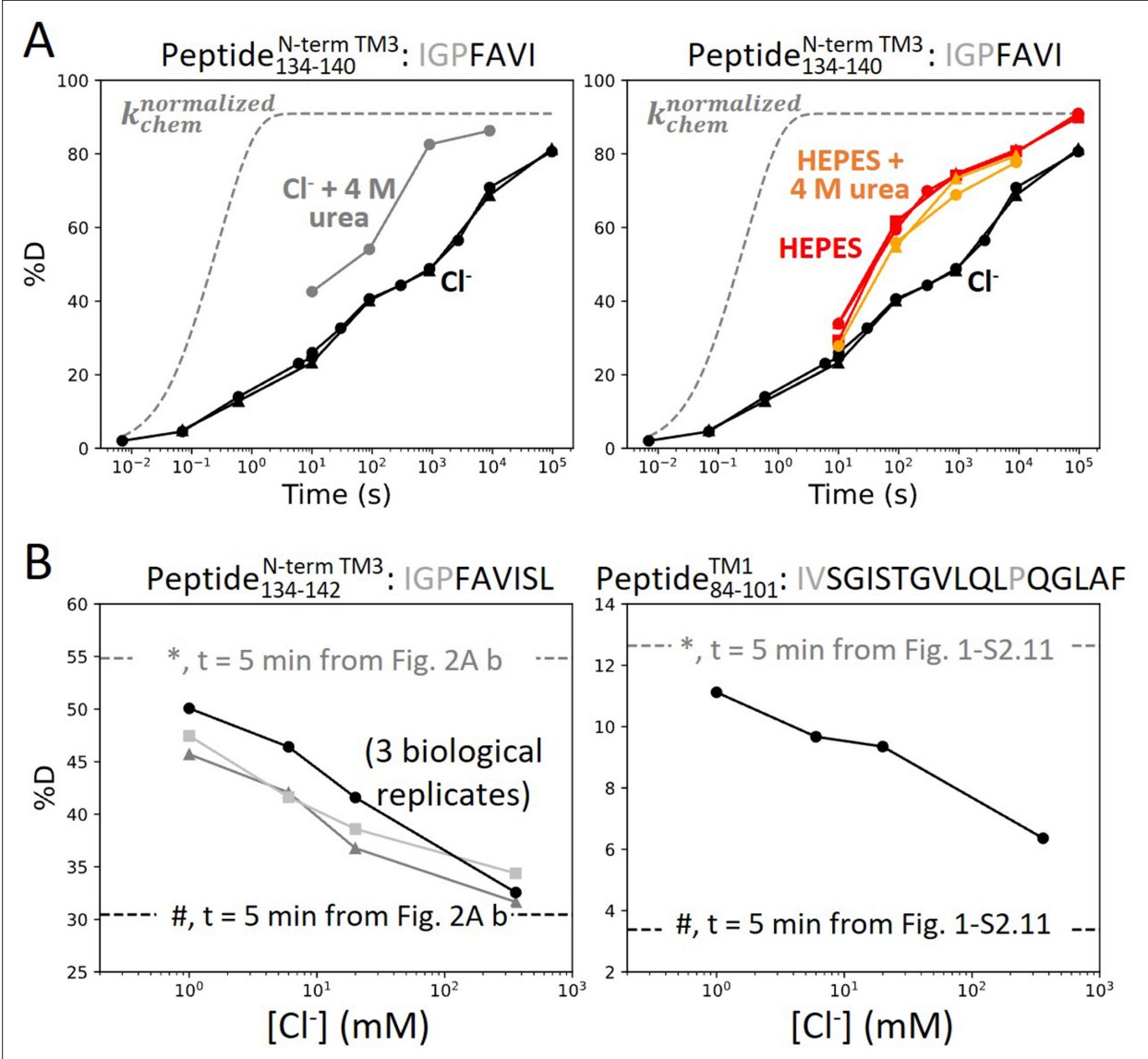

**Figure 3.** Anion binding folds and stabilizes prestin's binding site. (**A**) Deuterium uptake plots for the N-terminus of TM3 (Peptide$_{134-140}$) measured in the absence and presence of 4 M urea, in a background of (left) Cl⁻ and (right) HEPES. Replicates (circles, triangles, and squares): 2 in Cl⁻, 3 in HEPES, 2 in HEPES with urea, biological. Gray dashed curves represent deuterium uptake with $k_{chem}$ (***Bai et al., 1993***; ***Nguyen et al., 2018***), normalized with the back-exchange level. (**B**) Deuteration levels for (left) the N-terminus of TM3 (Peptide$_{134-142}$) in three biological replicates and for (right) TM1 (Peptide$_{84-101}$) after 5 min labeling upon titrating Cl⁻ to apo state of prestin. Dashed lines indicate deuteration levels at $t$ = 5 min (* and # for apo and Cl⁻-bound states, respectively) taken from ***Figure 2A.b*** and ***Figure 1—figure supplement 2.11***. Residues in gray denoted in the peptide sequence do not contribute to the deuterium uptake curve.

presence of salicylate, HDX slowed across the TMD, with the greatest effect seen at the anion-binding site (10-fold; $\Delta\Delta G^{\text{bindingsite}}_{\text{salicylate}-\text{Cl}-}$ = 1.4 kcal/mol) (***Figure 4***), indicating that salicylate binding to prestin globally stabilized the TMD, primarily at the anion-binding site. These stability changes provide a thermodynamic context to the cryo-EM structures (***Bavi et al., 2021***).

## Prestin in a lipid bilayer exhibits a highly dynamic TM6

We chose to measure the HDX of prestin in GDN micelle to match the cryo-EM conditions (***Bavi et al., 2021***; ***Ge et al., 2021***; ***Butan et al., 2022***). Structures of GDN-solubilized prestin in Cl⁻ obtained by three research groups are indistinguishable (Cα RMSD <1 Å), demonstrating the reproducibility and robustness of the system.

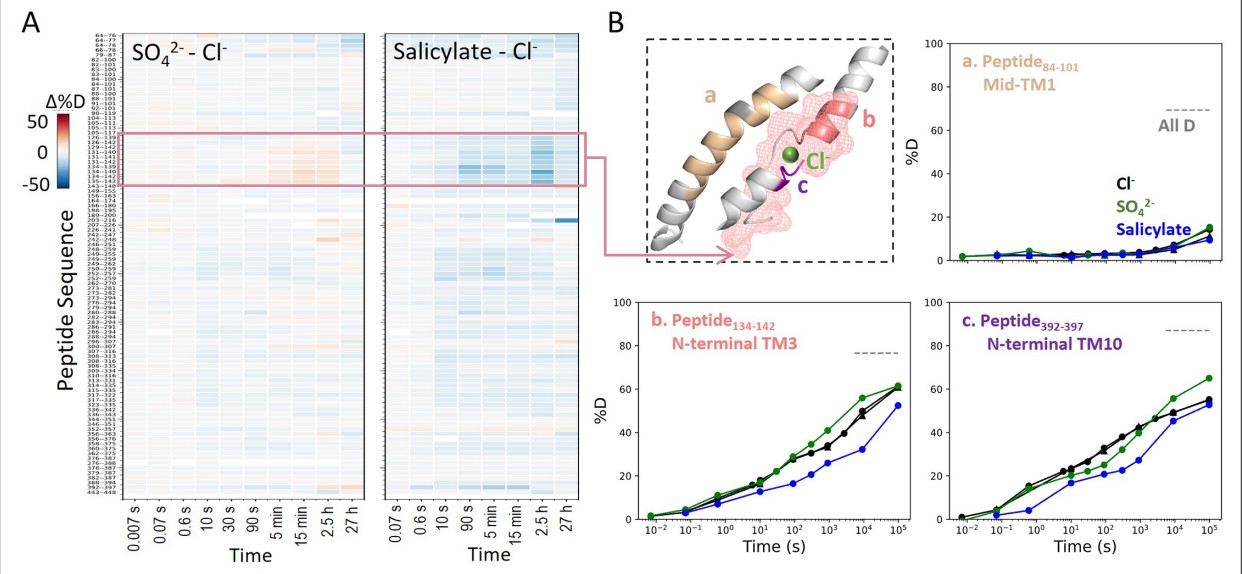

**Figure 4.** Prestin's dynamics are regulated by anions of varying identities. (**A**) Heatmaps showing the difference in deuteration levels at each labeling time for all transmembrane domain (TMD) peptides measured in $SO_4^{2-}$ or salicylate compared to $Cl^-$. Peptide sequences are displayed on the $y$-scale and legible through the high-resolution image. (**B**) The structure shows the anion-binding pocket with the putative position of the bound $Cl^-$. The pink mesh highlights the region with the greatest HDX response to binding to various anions. Colored regions correspond to peptides whose deuterium uptake plots are shown when the protein is in $Cl^-$ (black, two biological replicates shown in circles and triangles), $SO_4^{2-}$ (green), and salicylate (blue). Prolines are colored in gray. Gray dashed lines indicate deuteration levels in the full-D control.

To evaluate the dynamics in a more native membrane environment, we measured HDX of prestin reconstituted in nanodisc (porcine brain total lipid extract). Except for TM6, HDX for prestin in nanodisc highly resembled that in micelles including the anion-binding pocket (*Figure 5*). Such high agreement between the folding stability in these two membrane mimetics suggest that our findings on prestin's anion-binding site and its folding equilibrium are pertinent to prestin residing in a lipid bilayer. This is not surprising because structures for human prestin in GDN and nanodisc are shown to be nearly identical (Cα RMSD = 0.2 Å) (*Ge et al., 2021*).

Interestingly, nanodisc-embedded prestin displayed a less stable TM6 for the intracellular portion than prestin in micelles, manifested by the 10-fold HDX increase (Region$_{275-282}$: Peptide$_{273-282}$ + 2 peptides) (*Figure 5*, *Figure 1—figure supplement 2.53–2.55*). TM6 defines the interface between prestin and the lipid bilayer, and has been proposed to mediate area expansion through helical bending (*Bavi et al., 2021*). The exact role of TM6 in regulating prestin's conformational cycle is currently under investigation.

## Incremental unfolding of prestin's binding site versus cooperative unfolding of the lipid-facing helices

Our broad HDX time range and dense sampling allowed us to observe effects at the residue level. In particular, the binding site of prestin (Region$_{128-140}$ and Region$_{394-397}$) exhibited a broad deuterium uptake curve in $Cl^-$, indicative of helix fraying where exchange of deuterium occurs from multiple states that differ by one hydrogen bond (*Figure 6A*). Such HDX pattern is consistent with the associated residues undergoing sequential unfolding with distinct PFs (i.e., stability). Site-resolved PFs obtained using PyHDX (*Smit et al., 2021*) support that the stability increased residue-by-residue for TM3 for amide protons located further away from the substrate (*Figure 6—figure supplement 1*). This gradual increase in residue stability along the helices is indicative of helix fraying starting from prestin's binding site.

In contrast, we observed much more cooperative unfolding in prestin's lipid-facing helices, with exchange occurring from one or a few high energy states where a set of hydrogen bonds are broken concertedly. Cooperatively exchanging residues have similar PFs and a characteristic sigmoidal

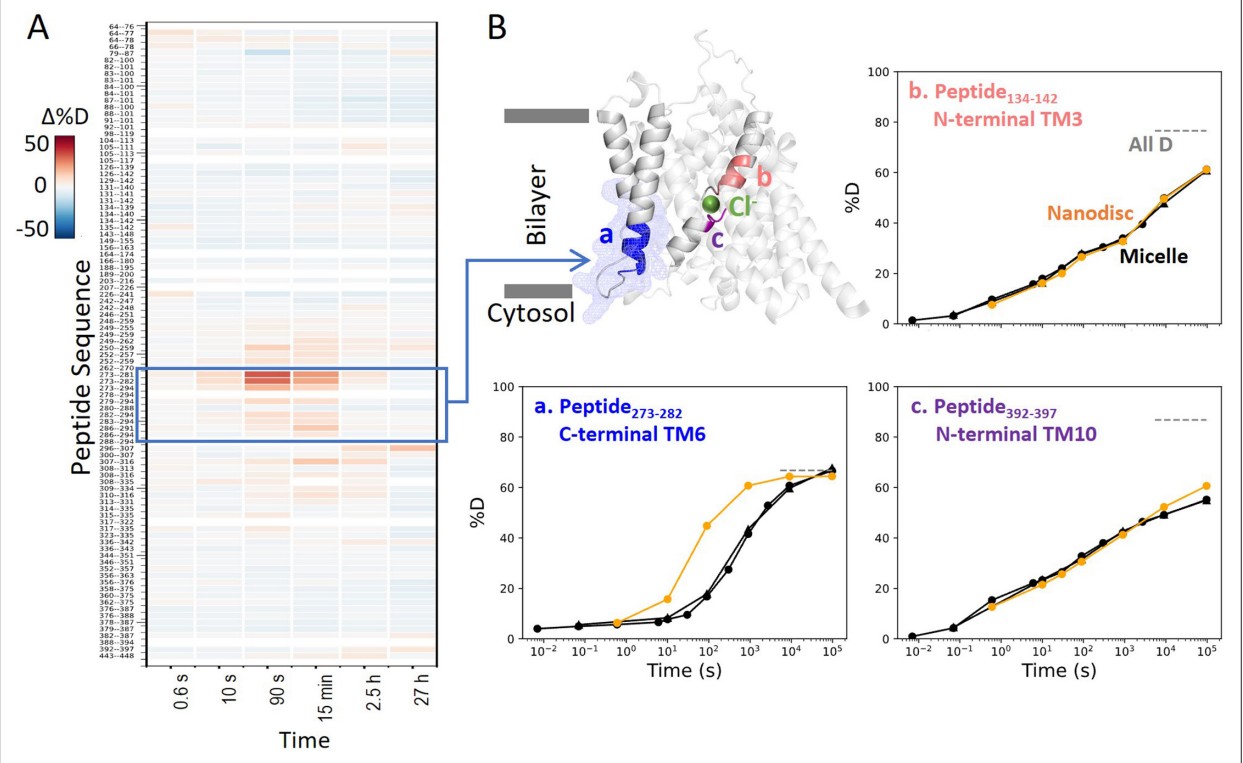

**Figure 5.** Prestin in nanodisc displays similar folding stability to prestin in micelle, except for a more dynamic TM6. (**A**) Heatmaps showing the difference in deuteration levels at each labeling time for all available transmembrane domain (TMD) peptides measured for prestin in nanodisc (porcine brain total lipid extract) compared to prestin in detergent micelle (glyco-diosgenin, GDN), both in Cl⁻. Peptide sequences are displayed on the y-scale and legible through the high-resolution image. (**B**) The structure shows the TMD for one subunit of prestin with the putative position of the bound Cl⁻. The blue mesh highlights the region where the greatest HDX difference was seen for prestin in nanodisc compared to micelle. Colored regions correspond to peptides whose deuterium uptake plots are shown when the protein is in micelle (black, biological duplicates shown in circles and triangles) and in nanodisc (orange). Gray dashed lines indicate deuteration levels in the full-D control.

deuterium uptake curve for the associated peptide, as seen in prestin's intracellular portion of TM6 (Region$_{275–282}$: Peptide$_{273–282}$ + 4 peptides) (*Figure 6A*, *Figure 1—figure supplement 2.53-2.57*).

To characterize the degree of cooperativity for the HDX at the N-terminus of TM3 (Peptide$_{134–140}$) and the intracellular portion of TM6 (Peptide$_{273–282}$), we fit the deuterium uptake curves as a sum of exponentials (*Hamuro, 2021*), $D(t) = \sum_{i=1}^{n} \left(1 - e^{-k_i t}\right)$, where $k_i$ is the exchange rate and $n$ is the number of exponentials, ranging from one to the number of exchange-competent residues. The value of $n$ was determined by the quality of the fit, evaluated by $\chi^2$ and having a relative error smaller than one (i.e., standard deviation for $k_i$ less than $k_i$ itself). HDX data for the N-terminus of prestin's TM3 (Peptide$_{134–140}$) were fit with four well-separated exponentials with rates spanning five log units for the four residues (*Figure 6A*). The need to individually fit each site indicates a lack of cooperativity and helix fraying. In contrast, the peptide representing the intracellular portion of TM6 (Peptide$_{273–282}$) has eight residues yet it could be fitted with only three rates spanning less than two log units (*Figure 6A*). This rather concerted deuterium uptake was independent of the anion substrate identity and also observed for TM1, TM5b, the intracellular portion of TM7, and the N-terminus of TM8 (*Figure 1—figure supplement 2.6-2.16, 2.42–2.43, 2.45–2.51, 2.58–2.63, 2.78–2.79*).

We define a parameter $\sigma_{\Delta G}$ to quantify the degree of folding cooperativity. The value of $\sigma_{\Delta G}$ is calculated as the standard deviation for the free energies of unfolding ($\Delta G$) for exchange-competent residues comprising the peptide. When a region folds 100% cooperatively, $\sigma_{\Delta G}$ is zero as all residues have the same $\Delta G$. As the diversity increases (lower cooperativity), the $\sigma_{\Delta G}$ value becomes larger. The accuracy of the $\Delta G$ determination at residue level can be increased by comparing uptake curves for overlapping peptides and/or deconvoluting isotope envelopes (*Hamuro, 2021*). Here, we assigned exchange rates ($k_i$), obtained from the fitting method mentioned above, to residues based on the

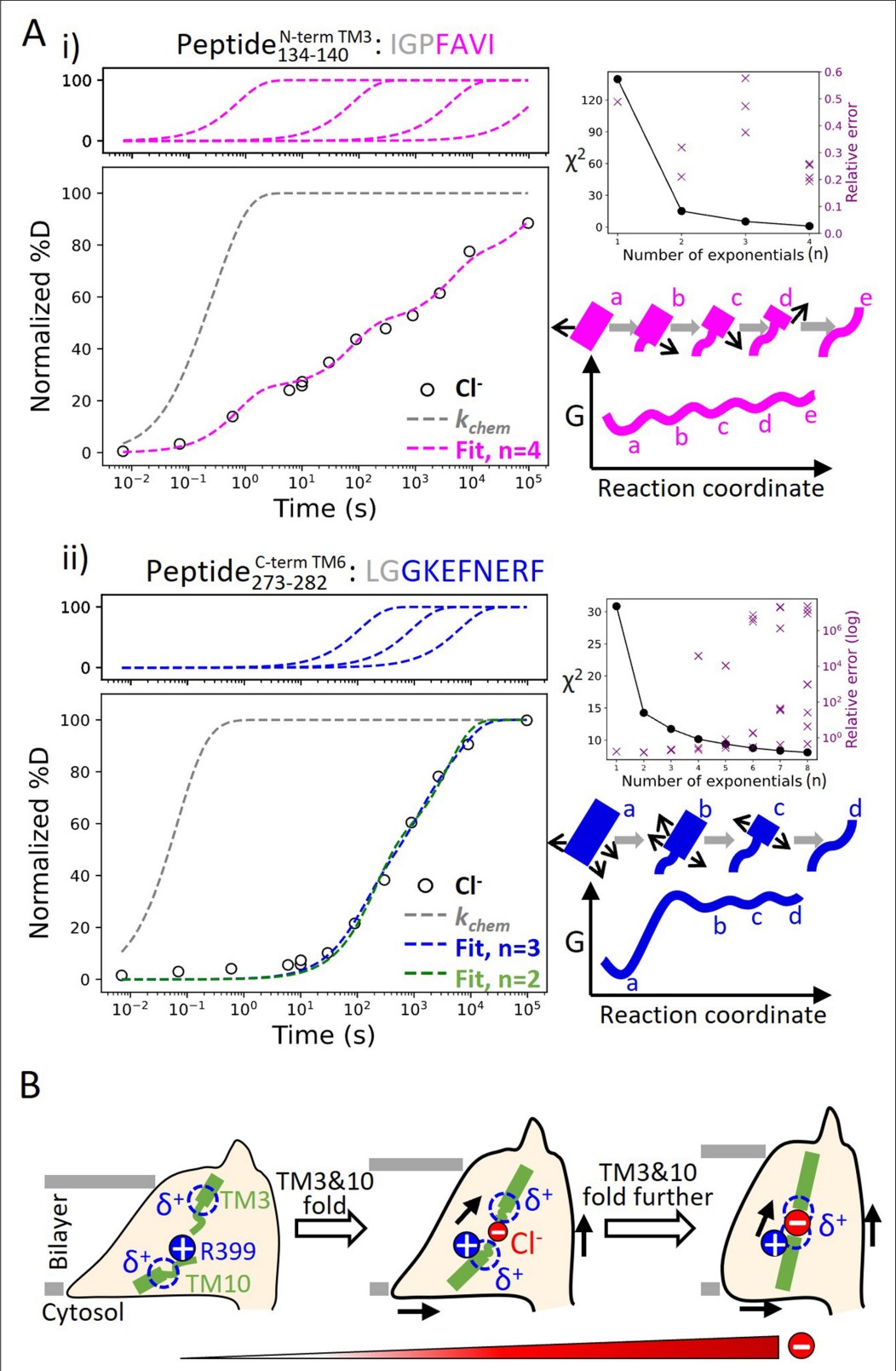

**Figure 6.** Helix folding cooperativity and the proposed mechanism for prestin's electromotility. (**A**) Left: Deuterium buildup curves for (i) the N-terminal TM3 (Peptide$_{134-140}$) and (ii) the intracellular portion of TM6 (Peptide$_{273-282}$) in Cl$^-$ depicting helix fraying and mild cooperativity, respectively. Circles: experimental deuteration levels, normalized with in- and back-exchange levels. Gray dashed curves: hypothetical intrinsic uptake curves (PF = 1). On the top

*Figure 6 continued on next page*

*Figure 6 continued*

shows individual exponentials whose sum is fitted to the experimental values and plotted on the main buildup curves. Residues in gray denoted in the peptide sequence do not contribute to the deuterium uptake curve. Upper right: $\chi^2$ and the relative error as the number of fit exponentials increases, used to assess the quality of fit. Lower right: Models and free energy surface of unfolding illustrating the difference between (i) fraying and (ii) mild cooperativity. (**B**) Mechanism for prestin's conformational transition from the expanded to the contracted state regulated by the anion concentration. Green rectangles and curved lines: folded and unfolded fractions, respectively, of TM3 and TM10. Blue filled circle: R399. Blue dashed circle: partial positive charges from TM3 and TM10 helical dipoles. Red filled circle: anions, with the size of the circle depicting anion concentrations. Black arrows: prestin's conformational change.

The online version of this article includes the following figure supplement(s) for figure 6:

**Figure supplement 1.** PyHDX fitting supports that prestin exhibits helix fraying at the N-terminus of TM3 and mild cooperativity at the intracellular portion of TM6.

---

directionality of helix fraying, with residues closer to the end of a helix having faster rates. When there is ambiguity on which rate to assign to a given residue, the geometric mean of the rates was used (Materials and methods). We found that prestin's intracellular portion of TM6 (Peptide$_{273-282}$) has $\sigma_{\Delta G}$ = 1.1, indicating mild cooperativity, whereas the non-cooperative N-terminal TM3 (Peptide$_{134-140}$) has a $\sigma_{\Delta G}$ = 2.9. This significant decrease in folding cooperativity for helices directly involved in the Cl$^-$-binding site likely has functional consequences related to prestin's electromotility, as discussed below.

## Discussion

Using HDX-MS, we provide novel information on the structural dynamics of prestin in its apo state, for which there is not an associated cryo-EM structure. We demonstrate that prestin displays very similar dynamics in nanodisc as in micelles, except for a destabilized lipid-facing helix TM6 that is critical for mechanical expansion. We have explored the energetic and conformational differences between prestin, a voltage-dependent motor, and its mammlian relative SLC26A9, a representative member of the SLC26 family of anion transporters for which a cryo-EM structure is available. Our data point to major differences in the energetics at the anion-binding site of prestin and SLC26A9 despite their structural similarities. This comparison addresses underlying mechanistic questions related to the unique properties of prestin, the origin of its voltage dependence, and the potential mechanisms that couple charge movements to electromotility.

We showed that prestin displays an unstable binding site, regardless of being in nanodisc or micelles (*Figure 5*). Upon Cl$^-$ unbinding, the binding site unfolds by one helical turn at the electrostatic gap formed by the abutting (antiparallel) short helices TM3 and TM10 (*Figures 2A and 3*). We measured an increase in local $\Delta\Delta G$ = 1.8–3.5 kcal/mol upon anion binding. This energy difference is within the range of the $\Delta\Delta G$ = 2.4 kcal/mol estimated from having a 60-fold excess of Cl$^-$ above the EC$_{50}$ (6 mM) (*Oliver et al., 2001*). Similar folding events upon anion binding are absent in SLC26A9 (*Figure 2B*), pointing to a key role of the bound anion as a structural element in prestin, stabilizing the natural repulsion between TM3- and TM10-positive helical macrodipoles. This phenomenon rationalizes the conundrum that prestin's voltage dependence requires the proximity of a bound anion to R399.

We find that anion binding to prestin mainly stabilizes the interface between the scaffold and the elevator domains (*Figure 1C*, *Figure 1—figure supplement 1A*). This phenomenon is consistent with an elevator-like mechanism during prestin's conformational transition from the expanded to the contracted state (*Bavi et al., 2021*). Anion binding stabilizes prestin's intracellular cavity and slightly destabilizes regions facing the extracellular matrix. This effect can result from changes in solvent exposure, as the intracellular water cavity may shrink as prestin contracts. For SLC26A9, the destabilization upon anion binding at the intracellular cavity likely results from a shift from the outward-facing state to the inward-facing state (*Figure 1C*, *Figure 1—figure supplement 1B*), supporting the alternate-access mechanism for this fast transporter (*Walter et al., 2019*; *Chi et al., 2020*). Similar HDX changes, that is, increased HDX on the intracellular side while decreased HDX on the extracellular side, have been observed in other transporters during their transition from outward- to inward-facing states (*Merkle et al., 2018*; *Martens et al., 2018*). Prestin's distinct HDX response compared

to a canonical transporter is consistent with it being an incomplete anion transporter (*Oliver et al., 2001*; *Schaechinger and Oliver, 2007*).

The HDX data for prestin in Cl⁻, $SO_4^{2-}$, and salicylate support an allosteric role for the anion binding at the TM3–TM10 electrostatic gap (*Rybalchenko and Santos-Sacchi, 2003*; *Rybalchenko and Santos-Sacchi, 2008*; *Song and Santos-Sacchi, 2010*; *Figure 4*). $SO_4^{2-}$ binding leads to shifts in the NLC toward positive potentials, thus stabilizing multiple conformations that are on average more expanded than prestin in Cl⁻ (*Bavi et al., 2021*; *Rybalchenko and Santos-Sacchi, 2008*; *Muallem and Ashmore, 2006*). Since the binding of $SO_4^{2-}$ to prestin is weaker than that of Cl⁻⁶, the slight increase in HDX at the binding site likely reflects more prestin molecules adopting the apo state. Salicylate binding inhibits prestin's NLC and yet the molecular basis of such inhibition remains obscure (*Bavi et al., 2021*; *Gorbunov et al., 2014*). *Bavi et al., 2021* showed that binding of salicylate occludes prestin's binding pocket from solvent and inhibits the movement of TM3 and TM10. This is fully consistent with the 10-fold HDX slowing found for the N-termini of TM3 and TM10 upon salicylate binding as compared to the rest of the protein. Our HDX data, together with results from *Bavi et al., 2021*, suggest that salicylate likely inhibits prestin's NLC by restricting the dynamics of the anion-binding site.

We identified helix fraying at the anion-binding site of prestin based on its broad deuterium uptake curve in the presence of Cl⁻, consistent with a multi-state landscape (*Figure 6A*). This fraying suggests that an increase in the anion concentration would promote helical propensity at TM3 and TM10, and is inconsistent with a cooperative (two-state) model involving an equilibrium between an apo state and a single bound state (*Figure 6A*). Therefore, we suggest that a two-state model of prestin's conformational changes with a high energy barrier would be insufficient to explain its fast kinetics, whereas charge movement is facilitated by crossing multiple shallow barriers (*Rybalchenko and Santos-Sacchi, 2008*; *Santos-Sacchi et al., 2009*).

We propose that having a Cl⁻-binding site that frays can promote prestin's fast motor response which is thought to have evolved independently of its voltage-sensing ability (*Tan et al., 2012*; *Tang et al., 2013*). While the stability for TM10 is similar for prestin and SLC26A9, the latter protein exhibits a more stable, non-fraying TM3 (*Figure 2*). Notably, the normally highly conserved Pro136 in mammalian prestin is replaced with a Threonine in SLC26A9 and other vertebrates that express non-electromotile prestin (*Figure 2—figure supplement 2*). This Pro136Thr substitution, based on our *Upside* MD simulations, results in a hyper-stabilized TM3 that would otherwise have similar folding stability as TM10 (*Figure 2—figure supplement 2*). A Pro136Thr mutation in rat prestin also leads to a shift of NLC toward depolarized potentials (*Schaechinger et al., 2011*). These thermodynamic and functional consequences of having a destabilized TM3 with Pro136, which now has similar stability as TM10, lead us to hypothesize that prestin's fast mechanical activity may be promoted by having simultaneous fraying of the TM3 and TM10 helices.

Helices that exhibit cooperative unfolding all appear to be lipid-facing helices, including TM6–TM7, TM1, TM5b, and TM8. The region with the most pronounced cooperativity, the intracellular portion of TM6, has a series of glycines including the consecutive G274–G275 pair that underlies the 'electromotility elbow', a helical bending contributing to the largest cross-sectional area (expanded conformation) and the thin notch in the micelle (*Bavi et al., 2021*). Importantly, HDX for prestin in nanodisc reveals a significant stability decrease at TM6 while the remaining regions retained similar dynamics as in micelles. This high sensitivity of TM6 stability to membrane environment, together with the structural consequences of cooperativity, speak to the its significance in prestin's area expansion. We propose that cooperativity allows for long-range allostery (*Hilser and Thompson, 2007*) so that the lipid-facing helices, particularly TM6, can adopt large-scale structural rearrangements as induced by voltage sensor movements, thereby achieving rapid electromechanical conversions of prestin. The exact mechanism through which cooperativity contributes to prestin's electromotility remains a key question.

Based on the structural and allosteric role of Cl⁻ binding at the TM3–TM10 electrostatic gap, we propose a model in which prestin's conformational changes and electromotility are regulated by the folding equilibrium of the anion-binding site (*Figure 6B*). In our model, anion binding participates in a local electrostatic balance that includes the positively charged R399 and the positive TM3–TM10 helical macrodipoles. In the apo state, the anion-binding site unfolds due to the electrostatic repulsion between these positively charged groups. Being a buried charge, R399 may exit from the electric field

concentrated in the lower dielectric environment of the protein, and move into the solvent region as it lacks a neutralizing anion. This event is coupled to the allosteric expansion of prestin's membrane footprint (*Bavi et al., 2021*). Anion binding partially neutralizes the positive electric field at the binding site, an event that is coupled to the residue-by-residue folding for the N-termini of TM3 and TM10 as well as the shortening of the electrostatic gap. This folding event results in a more focused electric field and consequent contraction of prestin's intermembrane cross-sectional area. At physiologically relevant low Cl⁻ concentration (1.5–4 mM) (*McPherson, 2018*), prestin's binding site is likely to be only partially folded. Complete folding may be achieved by membrane potential acting on the TM3–TM10 helical dipoles, which leads to the movement of the voltage sensor across the electric field and the rapid areal expansion for the TMDs (i.e., electromotility).

## Structure of prestin in HEPES and low Cl⁻ levels

We associate the HEPES solvent condition to an apo state as the HEPES anion is too big to fit into the chloride site and prestin has a right-shifted NLC (*Rybalchenko and Santos-Sacchi, 2008*), To investigate further whether HEPES anion binds, we determined the structure of prestin in the HEPES-based buffer using cryo-EM. Our initial attempt to solve the structure in the complete absence of Cl⁻ was unsuccessful due to sample aggregation under cryogenic conditions. Aggregated particles were greatly reduced in the presence of 1 mM Cl⁻, and we were able to solve the structure of GDN-solubilized prestin in 1 mM Cl⁻, containing 190 mM HEPES, at a nominal resolution of 3.4 Å (Appendix 3, Appendix 3—figures 1 and 2, *Appendix 3—table 1*). Surprisingly, under these conditions prestin adopted a 'compact' conformation, virtually identical to the previously reported Cl⁻-bound 'Up' state (*Bavi et al., 2021*), displaying a clear density at the anion-binding site (*Appendix 3—figure 1*). This density is incompatible with the placing of a HEPES molecule, and we reason it to be a small population of Cl⁻-bound prestin resulting from a weak Cl⁻ affinity (e.g., $EC_{50}$ = 6 mM[11] implies 17% bound). This result is consistent with the fundamental role of bound anions in the conformational stability of prestin and supports a new role for the folding equilibrium of the anion-binding site in the mechanism of voltage sensing. Ultimately, however, understanding the underlying mechanism for prestin's electromotility necessitates consideration of physiological elements such as membrane potential, kHz frequency, and protein–lipid interactions.

## Conclusions

We applied HDX-MS to the study of prestin's electromotility and identified folding events that are likely critical for function but had escaped detection by cryo-EM. The folding equilibrium of the Cl⁻-binding site and its dependence on Cl⁻ concentration appears to rationalize the conundrum of how an anion that binds in proximity to a positive charge (R399), can enable the NLC of prestin. We directly compared the dynamics of prestin in nanodisc to in micelles and identified TM6 as a potential mechano-sensing helix. We observed fraying of the helices forming the anion-binding site, which contrasts with cooperative unfolding of the lipid-facing helices. We believe that the non-cooperative fraying of the helices involved in voltage sensing may allow for fast charge movements within the electric field. This heightened sensitivity of the voltage sensor then induces large-scale motions of the lipid-facing helices, enabled by their cooperativity (or allostery), thereby altering the cross-sectional area of prestin. These principles warrant further investigation.

## Materials and methods
### Sample preparation for prestin and SLC26A9

Generation of the dolphin prestin and mouse SLC26A9 constructs, protein overexpression, and purification were conducted using the same protocol as previously described (*Bavi et al., 2021*). The HEK293S GnTI⁻ cells in suspension that were used for protein expression and purification were obtained and authenticated from ATCC (CRL-3022). Following the cleavage of the GFP tag, the protein was purified by size-exclusion chromatography (SEC) on a Superose 6, 10/300 GE column (GE Healthcare), with the running buffer being either the 'Cl⁻/H₂O Buffer' or the 'SO₄²⁻/H₂O Buffer', including 1 μg/ml aprotinin and 1 μg/ml pepstatin (*Bavi et al., 2021*). The 'Cl⁻/H₂O Buffer' contained 360 mM NaCl, 20 mM Tris, 3 mM dithiothreitol (DTT), 1 mM Ethylenediaminetetraacetic acid (EDTA), and 0.02% GDN at pH 7.5. The 'SO₄²⁻/H₂O Buffer' contained 140 mM Na₂SO₄, 5 mM MgSO₄, 20 mM Tris, 0.02% GDN,

and 10–15 mM methanesulfonic acid to adjust the pH to 7.5. Peak fractions containing the sample were concentrated on a 100 K MWCO centrifugal filter (Millipore) to 2–3 mg/ml, flash-frozen in liquid nitrogen, and kept at −80°C until use.

For prestin reconstitution into nanodiscs, porcine brain lipid extract (Avanti) was first solubilized in 'Cl⁻/H₂O Buffer' supplemented with 3% GDN to a 10 mg/ml concentration through successive rounds of sonication and freeze–thaw cycles. MSP1E3D1 was purified as previously described (*Alvarez et al., 2010.Clark et al., 2022*). Reconstitution was performed using the 'on-column' method (*Clark et al., 2022*) using a 1:5:600 (prestin:MSP:lipid) molar ratio, with the final buffer consisting of 20 mM Tris–HCl and 150 mM NaCl at pH 7.5.

For structure determination by cryo-EM, prestin purification was performed in buffers containing 360 mM NaCl as previously described (*Bavi et al., 2021*), except that the SEC running buffer now consisted of 190 mM HEPES, ~95 mM Tris-base (used to adjust the pH to 7.5), 1 mM NaCl, 3 mM Dithiothreitol (DTT), 0.02% GDN, 1 µg/ml aprotinin, and 1 µg/ml pepstatin. Peak fractions containing the sample were concentrated to 3 mg/ml and used immediately for grid preparation.

## Hydrogen–deuterium exchange

*Table 1* provides biochemical and statistical details for HDX in this study per recommendations by *Masson et al., 2019*. HDX reactions, quench, and injection were all performed manually. Prior to HDX, proteins purified in Cl⁻ and SO₄²⁻ were buffer exchanged to the same H₂O buffer without protease inhibitor using 7 K MWCO Zeba spin desalting columns (Thermo 89882). For HDX conducted in HEPES, proteins purified in Cl⁻ were dialyzed against the 'HEPES/H₂O Buffer' (150 mM HEPES, 0.02% GDN, pH adjusted to 7.5 by HEPES acid or base) using a 10 K MWCO dialysis device (Thermo Slide-A-Lyzer MINI 69570) with three times of buffer exchange for near-complete Cl⁻ removal. HDX in a solution of 93% deuterium (D) content was initiated by diluting 2 µl of 25 µM prestin or SLC26A9 stock in an H₂O buffer into 28 µl of the corresponding buffer made with D₂O (99.9% D, Sigma-Aldrich 151882). HDX was conducted in one of the two conditions: (1) $pD_{read}$ 7.1, 25°C; (2) $pD_{read}$ 6.1, 0°C. The D₂O buffers contained the same compositions as the corresponding H₂O buffers, except that Tris was replaced with phosphate for the 'SO₄²⁻/D₂O Buffer' for both HDX conditions, and the 'Cl⁻/D₂O Buffer' for HDX performed at $pD_{read}$ 6.1, 0°C. The $pD_{read}$ was adjusted to 7.1 or 6.1 by DCl for the 'Cl⁻/D₂O Buffers' and by NaOD for other D₂O buffers. For HDX in the presence of salicylate, 50 mM salicylate acid was added to the 'SO₄²⁻/D₂O Buffer' and the $pD_{read}$ was adjusted by NaOD. For HDX in the presence of urea, 4 M urea was added to the 'Cl⁻/D₂O Buffer' or the 'HEPES/D₂O Buffer', with accurate urea concentration determined to be 4.16 and 4.54 M, respectively, by refractive index using a refractometer (WAY Abbe) (*Warren and Gordon, 1966*).

HDX was quenched at various times, ranging from 1 s to 27 hr, by the addition of 30 µl of ice-chilled quench buffer containing 600 mM glycine, 8 M urea, pH 2.5. For HDX in the presence of urea, urea concentration in the quench buffer was adjusted to reach a 4 M final concentration. For HDX on prestin in nanodisc, the quench buffer also included 3 µl of 0.8% GDN and 3 µl of 300 mg/ml aqueous suspension of ZrO₂-coated silica (Sigma-Aldrich, reference no. 55261-U). The resulting mixture was incubated on ice for 1 min to remove lipids and solubilize prestin in GDN before being filtered through a cellulose acetate spin cup (Thermo Pierce, Waltham, MA, reference no. 69702) by centrifugation for 30 s at 13,000 × *g*, 2°C. Quenched reactions were immediately injected into a valve system maintained at 5°C (Trajan LEAP). Non-deuterated controls and MS/MS runs for peptide assignment were performed with the same protocol as above except D₂O buffers were replaced by H₂O buffers, followed by the immediate addition of the quench buffer and injection.

HDX reactions were performed in random order. No peptide carryover was observed as accessed by following sample runs with injections of quench buffer containing 4 M urea and 0.01% GDN. In-exchange controls accounting for forward deuteration toward 41.5% D in the quenched reaction were performed by mixing D₂O buffer and ice-chilled quench buffer prior to the addition of the protein. Maximally labeled controls (All D) accounting for back-exchange were performed by a 48-hr incubation with the 'Cl⁻/D₂O Buffer' at 37°C, followed by a 30-min incubation with 8 M of deuterated urea at 25°C.

## Protease digestion and liquid chromatography–mass spectrometry

Upon injection, the protein was digested online by a pepsin/FPXIII (Sigma-Aldrich P6887/P2143) mixed protease column maintained at 20°C. Protease columns were prepared in-house by coupling the protease to a resin (Thermo Scientific POROS 20 Al aldehyde activated resin 1602906) and hand-packing into a column (2 mm ID × 2 cm, IDEX C-130B). After digestion, peptides were desalted by flowing across a hand-packed trap column (Thermo Scientific POROS R2 reversed-phase resin 1112906, 1 mm ID × 2 cm, IDEX C-128) at 5°C. The total time for digestion and desalting was 2.5 min at 100 µl/min of 0.1% formic acid at pH 2.5. Peptides were then separated on a C18 analytical column (TARGA, Higgins Analytical, TS-05M5-C183, 50 × 0.5 mm, 3 µm particle size) via a 14 min, 10–60% (vol/vol) acetonitrile (0.1% formic acid) gradient applied by a Dionex UltiMate-3000 pump. Eluted peptides were analyzed by a Thermo Q Exactive mass spectrometer. MS data collection, peptide assignments by SearchGUI version 4.0.25, and HDX data processing by HDExaminer 3.1 (Sierra Analytics) were performed as previously described (*Zmyslowski et al., 2022*; *Lin et al., 2022*).

## HDX data presentation, quantification, and statistics

In our labeling convention, we name capitalized regions according to the third residue of each peptide since the first two residues have much faster $k_{chem}$ (*Bai et al., 1993*; *Nguyen et al., 2018*) and hence, exhibit complete back-exchange. Labeling times for HDX performed in $pD_{read}$ 6.1, 0°C were corrected to those in $pD_{read}$ 7.1, 25°C by a factor of 140, determined by the ratio of the $k_{chem}$ for full-length proteins in the two conditions – prestin: $\tau_{chem}^{pDread\ 6.1,\ 0\ ^\circ C} = 10$ s, $\tau_{chem}^{pDread\ 7.1,\ 25\ ^\circ C} = 74$ ms; SLC26A9: $\tau_{chem}^{pDread\ 6.1,\ 0\ ^\circ C} = 9$ s, $\tau_{chem}^{pDread\ 7.1,\ 25\ ^\circ C} = 65$ ms. Deuteration levels were adjusted with the 93% D content but not with back-exchange levels except for *Figure 6A*. For HDX in the presence of urea (*Figure 3A*), D contents of 76% and 74% were used to account for the volumes of 4.16 and 4.54 M urea in the 'Cl⁻/ $D_2O$ Buffer' and the 'HEPES/$D_2O$ Buffer', respectively.

HDX rates, PFs, and changes of free energy of unfolding ($\Delta\Delta G$s) in Results were estimated by fitting uptake curves of each peptide, after correcting for back-exchange levels, with a stretched exponential as described by *Hamuro, 2021*, except for discussion relevant to *Figure 6A*. Peptides with less than 10% D at the longest labeling time (~$10^5$ s) were not used for fitting. The residue-level $\Delta\Delta G$ values presented in the full-length proteins in *Figure 1C* were estimated by averaging $\Delta\Delta G$ values for peptides containing the residue. We note that this stretched exponential method is only a rough approximation to extract $\Delta\Delta G$s and our major conclusions are not dependent on this fitting method.

For fitting HDX data and extracting rates relevant to *Figure 6A*, HDX data were first normalized to in- and back-exchange levels. Given the helices are likely to exchange by fraying, the exchange rate for each residue was assigned based on their distance from the end of the helix. When a residue could not be assigned to a single rate, the geometric mean of the possible rates was used, for example, the HDX data for the 8-residue Peptide$_{273–282}$ were well fit with three exponentials, and the three associated rates, $k_1$, $k_2$, and $k_3$, were assigned to the 8 residues according to $k_1$, $k_1$, $(k_1 k_1 k_2)^{1/3}$, $k_2$, $k_2$, $(k_2 k_2 k_3)^{1/3}$, $k_3$, and $k_3$. These rates were used to calculate folding stability according to $\Delta G = -RT \ln \left( k_{chem}/k_i - 1 \right)$.

A hybrid statistical analysis used to generate *Figure 1—figure supplement 1* was performed as described by *Hageman and Weis, 2019*, with significance limits defined at $\alpha = 0.05$.

## MD simulations

Simulations were conducted in our Upside molecular dynamics package (*Jumper et al., 2018a*; *Jumper et al., 2018b*) using a membrane thickness of 38 Å. Missing residues for prestin (PDB 7S8X) were built using MODELLER (*Sali and Blundell, 1993*) and the placement within the bilayers was accomplished using Positioning of Proteins in Membranes webserver (*Lomize et al., 2022*). Local restraints in the form of small springs between nearby residues were used to maintain the native structure of cytosolic domains, as well as to the TM13–TM14 helices. Also, the distance between the two TMDs was held fixed. We ran 28 temperature replicas between 318 and 360 K.

## Cryo-EM sample preparation and data collection

The cryo-EM data were collected from three separate samples and microscope sessions. For dataset 1, 3.5 µl of prestin sample was applied to Quantifoil 200-mesh 1.2/1.3 Cu grids (Quantifoil) that were plasma cleaned for 30 s at 20 W. For datasets 2 and 3, 3.5 µl of prestin sample was applied to UltrAuFoil 300-mesh 1.2/1.3 grids (Quantifoil UltrAuFoil) that were plasma cleaned for 40 s at 20 W.

The remaining sample preparation and imaging conditions were kept constant for all three samples. Grids were blotted at 22°C and 100% humidity with a blot time of 3.5 s and a blot force of 1, and flash-frozen into liquid ethane using a Vitrobot Mark IV (Thermo Fisher). Grids were imaged at The University of Chicago Advanced Electron Microscopy Facility on a 300 kV Titan Krios G3i electron microscope equipped with a Gatan K3 camera in CDS mode, a GIF energy filter (set to 20 eV) and with magnification set to ×81,000, corresponding to a physical pixel size of 1.068 Å. Movies were acquired at a dose of 1.2 $e^-/Å^2$ for 50 frames (corresponding to a total dose of 60 $e^-/Å^2$) and a defocus range of −0.7 to −2.1 μm. 2153 movies were collected for dataset 1, 1928 movies for dataset 2, and 6665 movies for dataset 3.

## Cryo-EM image processing

Cryo-EM data processing was performed using Relion-4.0 (*Kimanius et al., 2021*) and cryoSPARC v4.1 (*Punjani et al., 2017*). Movies from the different datasets were motion-corrected independently in Relion with a bin-1 pixel size of 1.068 Å. The motion-corrected micrographs were then combined and imported into cryoSPARC for Patch CTF Estimation. Unless otherwise mentioned, the following steps were performed in cryoSPARC. Particles were picked using template-based particle picking. The initial picks were curated using Inspect Picks and extracted at a box size of 256. Two rounds of 2D Classification were performed to filter out additional 'junk' particles, resulting in a set of 328,033 particles. Two consecutive rounds of ab initio reconstruction, followed by heterogeneous refinement with C1 symmetry were performed to obtain a stack of 170,313 particles. These particles were used as input for an initial round of non-uniform and local CTF refinement using C2 symmetry and exported to Relion for Bayesian Polishing. Additional 3D Classification was performed in Relion to remove additional 'junk' particles, resulting in a set of 170,313 particles. However, no additional classes were found. The final set of polished particles was then imported back into cryoSPARC and subjected to a final round of non-uniform and CTF refinement, resulting in a map with a nominal resolution of 3.4 Å, according to the gold-standard 0.143 Fourier shell correlation (FSC) criterion (*Rosenthal and Henderson, 2003*). Local resolution was estimated using local resolution estimation in cryoSPARC.

## Model building and refinement

A previous model of dolphin prestin (PDB 7S8X) was roughly fit into the density map and used as a template for model building. Initially, only a monomer was considered for model building, and the fitting was improved by running Phenix real space refinement (*Afonine et al., 2018*) with secondary structure restraints, morphing, and simulated annealing enabled. Subsequently, the monomer model was iteratively refined by manual inspection in Coot (*Brown et al., 2015*) and real space refinement without morphing and simulated annealing in Phenix. After several rounds of refinement, the second monomer was added using the apply_ncs tool in Phenix with C2 symmetry, and the resulting dimer model was subjected to an additional round of manual inspection in Coot. A chloride ion was manually added into the density observed in the anion-binding pocket in Coot. All figures related to the cryo-EM structure were prepared using UCSF ChimeraX (*Pettersen et al., 2021*).

## Additional information

### Funding

| Funder | Grant reference number | Author |
| --- | --- | --- |
| National Science Foundation | MCB 2023077 | Tobin R Sosnick |
| National Institute of General Medical Sciences | 1R35GM148233 | Tobin R Sosnick |
| National Institute on Deafness and Other Communication Disorders | R01 DC019833 | Eduardo Perozo |

The funders had no role in study design, data collection, and interpretation, or the decision to submit the work for publication.

## Author contributions
Xiaoxuan Lin, Conceptualization, Data curation, Formal analysis, Validation, Investigation, Visualization, Methodology, Writing - original draft, Writing – review and editing; Patrick R Haller, Data curation, Formal analysis, Validation, Investigation, Visualization, Methodology, Writing – review and editing; Navid Bavi, Validation, Investigation, Methodology, Writing – review and editing; Nabil Faruk, Validation, Investigation, Visualization, Methodology, Writing – review and editing; Eduardo Perozo, Conceptualization, Resources, Software, Formal analysis, Supervision, Funding acquisition, Validation, Visualization, Methodology, Project administration, Writing – review and editing; Tobin R Sosnick, Conceptualization, Resources, Data curation, Software, Formal analysis, Supervision, Funding acquisition, Validation, Visualization, Methodology, Project administration, Writing – review and editing

## Author ORCIDs
Xiaoxuan Lin ⬤ http://orcid.org/0000-0001-5356-9135
Tobin R Sosnick ⬤ https://orcid.org/0000-0002-2871-7244

Reviewer #1 (Public Review): https://doi.org/10.7554/eLife.89635.3.sa1
Reviewer #2 (Public Review): https://doi.org/10.7554/eLife.89635.3.sa2
Reviewer #3 (Public Review): https://doi.org/10.7554/eLife.89635.3.sa3
Author Response https://doi.org/10.7554/eLife.89635.3.sa4

# Additional files

## Supplementary files
• MDAR checklist

## Data availability
The raw mass spectrometry proteomics data have been deposited to the ProteomeXchange Consortium via the PRIDE partner repository with the dataset identifier PXD046965. The atomic structure coordinates have been deposited at the RCSB PDB under accession number 8UC1; and the EM maps have been deposited in the Electron Microscopy Data Bank under accession number EMD-42112. All materials generated during the current study are available from the corresponding author under a materials transfer agreement with The University of Chicago.

The following datasets were generated:

| Author(s) | Year | Dataset title | Dataset URL | Database and Identifier |
|---|---|---|---|---|
| Lin X, Sosnick TR | 2023 | Folding of prestin's anion-binding site and the mechanism of outer hair cell electromotility | https://www.ebi.ac.uk/pride/archive/projects/PXD046965 | PRIDE, PXD046965 |
| Haller P, Bavi N, Perozo E | 2023 | Cryo-EM structure of dolphin Prestin in low Cl- buffer | https://www.rcsb.org/structure/8UC1 | RCSB Protein Data Bank, 8UC1 |
| Haller P, Bavi N, Perozo E | 2023 | Cryo-EM structure of dolphin Prestin in low Cl- buffer | https://www.ebi.ac.uk/emdb/EMD-42112 | Electron Microscopy Data Bank, EMD-42112 |

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

# Appendix 1

## Heterogeneity and HDX kinetics

HDX in our study occurred mostly via EX2 kinetics where the observed exchange rate ($k_{ex}$) reports on the equilibrium (i.e., stability) rather than the opening rates of the exchange-competent states. The identification of EX2 behavior is supported by (1) $k_{ex}$ of prestin in the two HDX conditions differed by 140-fold, which can be attributed solely to the effect of pH and temperature on $k_{chem}$ for residues across the entire protein (*Appendix 1—figure 1A*); (2) the continuous shifts in the single isotopic envelopes toward higher $m/z$ with exchange time (*Appendix 1—figure 1*); (3) the envelopes had the binomial distribution expected when each site exchanges independently, supported by HDExaminer (v3.3) fits. The tracking of $k_{ex}$ with $k_{chem}$ also argues that prestin exhibits similar dynamics under the two HDX conditions, allowing us to combine the experiments after correcting for the difference in $k_{chem}$.

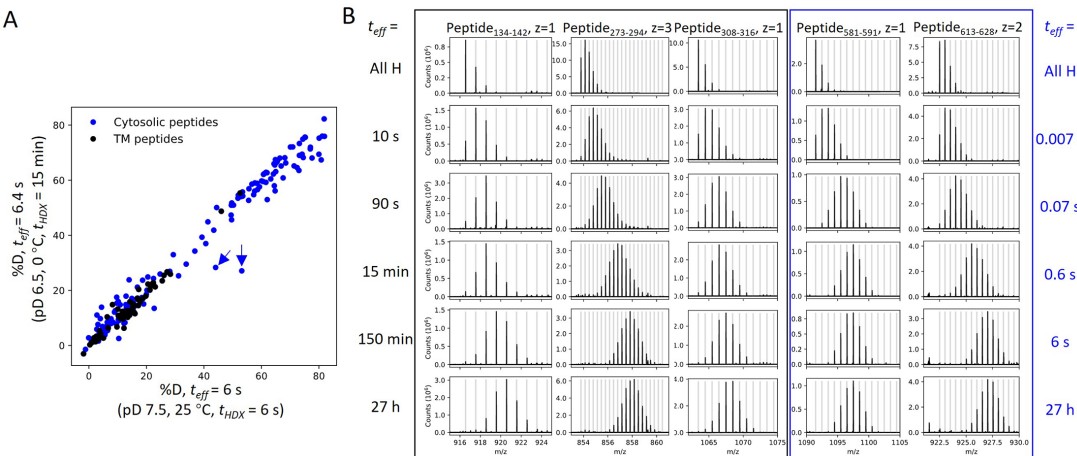

**Appendix 1—figure 1.** HDX for prestin occurs via EX2 mechanism. (**A**) Comparison of deuteration levels of all prestin peptides labeled under two HDX conditions: (1) $pD_{read}$ 7.1, 25°C, $t_{HDX}$ = 6 s; (2) $pD_{read}$ 6.1, 0°C, $t_{HDX}$ = 15 min, $t_{eff}$ = 6.4 s, where $t_{eff}$ represents effective HDX labeling time in $pD_{read}$ 7.1, 25°C (Materials and methods). Black: transmembrane (TM) peptides, blue: cytosolic peptides. The two outliers denoted by the blue arrows represent peptides covering the Cα2 helix in the STAS domain, whose large apparent %$D$ difference between the two conditions results from pH-/temperature-dependent dynamics (data not shown). (**B**) Mass spectra of a representative set of peptides showing progressive unimodal isotope envelopes toward high $m/z$ over time. Gray horizontal bars indicate theoretical $m/z$ values for the corresponding isotopes.

We observed bimodal isotopic envelopes for peptides in TM1 (Region$_{84-101}$: 9 peptides), with both envelopes increasing in mass over time, one exchanging slower than the other (*Appendix 1—figure 2A*). Bimodality can result from HDX occurring via EX1 kinetics where every opening event results in exchange; this occurs when the rate of reforming the hydrogen bond, $k_{close}$, is much slower than $k_{chem}$. The signature of EX1 kinetics is a decrease in the amplitude of the lighter envelope and a commensurate increase in the heavier amplitude over time (*Weis et al., 2006*). This EX1 behavior is observed in TM9 (Region$_{378-387}$) (*Appendix 1—figure 2*). Alternatively, bimodality can reflect the presence of two non- or slowly interconverting, structurally distinct populations, each having its own exchange behavior. Peptides in TM1 retained a 1:3 ratio of relative intensity for the heavy-to-light envelopes regardless of biological replicates or anionic conditions, pointing to kinetically distinct populations. In addition, both populations in TM1 exchanged via EX2 kinetics (*Appendix 1—figure 2*). Therefore, conformational heterogeneity best explains the exchange behavior of TM1 with an interconversion time between the two populations being longer than our longest labeling time (27 hr). We associate the slow population with the natively folded TM1, as observed in cryo-EM studies (*Bavi et al., 2021*), and focus on this population in the present study.

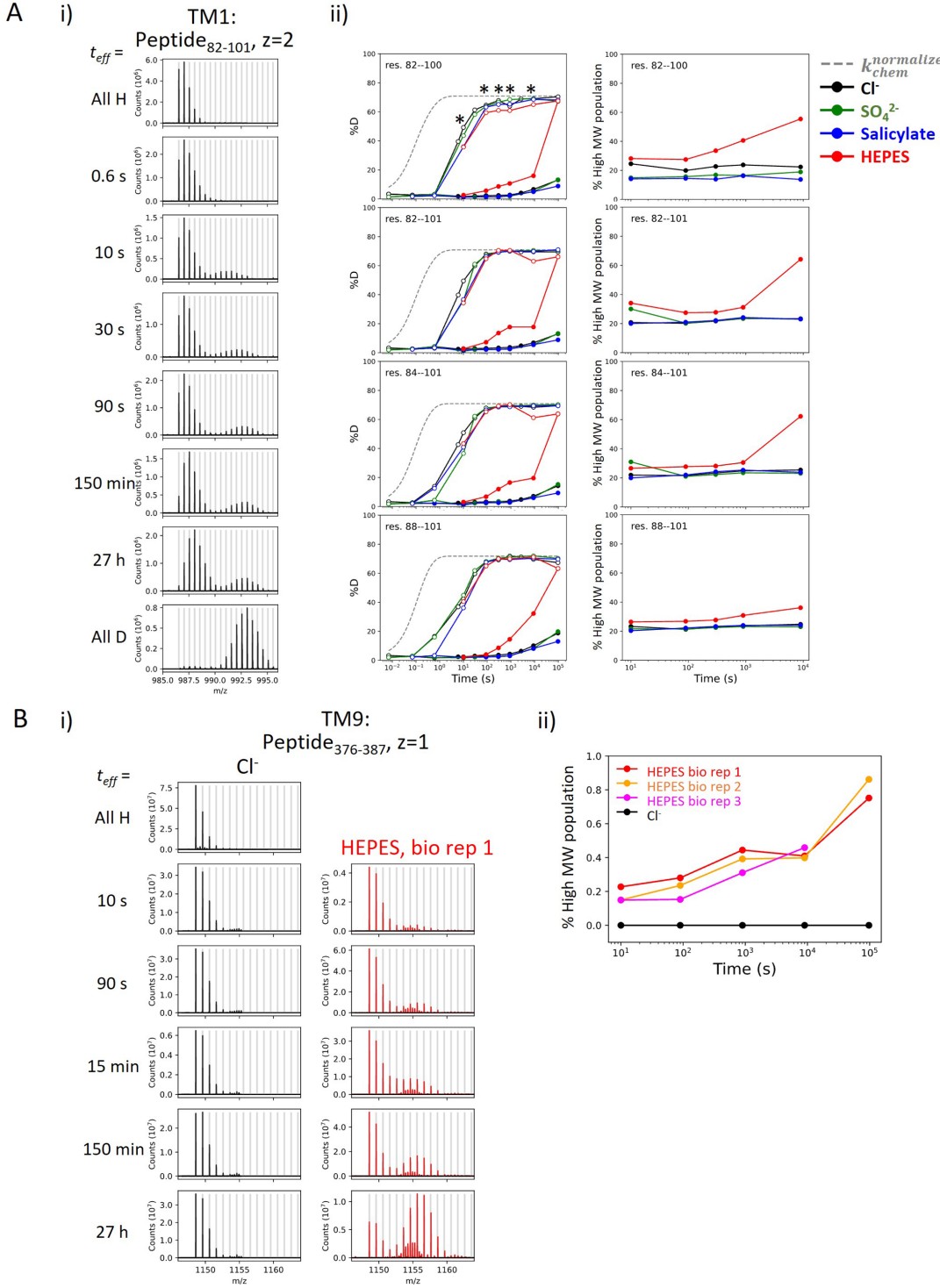

**Appendix 1—figure 2.** Heterogeneity and HDX kinetics in TM1 and TM9. (**A**) Conformational heterogeneity at TM1. (i) Example mass spectra for a TM1 peptide showing bimodal isotope envelopes, with both isotope distributions increasing in mass over time and exchanging via EX2 kinetics. (ii) Left: Deuterium uptake curves of example TM1 peptides plotting both the left (filled markers) and the right (empty markers) isotope envelopes under different anion conditions. Gray dashed curves represent deuterium uptake with $k_{chem}$, normalized with the in- and back-exchange levels. Only one replicate is shown for clarity. Asterisks represent time points used for the (right) population fraction analysis, chosen as two isotope envelopes are well separated. Fractions of the heavier envelope (i.e., % high MW population) in HEPES are higher than those in other conditions as the left envelope merges into the right envelope, resulting in less distinct separation between the two envelopes. (**B**) HDX for

*Appendix 1—figure 2 continued on next page*

*Appendix 1—figure 2 continued*

prestin's TM9 exhibits EX1 kinetics in the apo state. (i) Example mass spectra for a TM9 peptide measured for prestin in Cl⁻ (black) and HEPES (red). Identification of EX1 kinetics in HEPES is supported by the presence of two distinct mass envelopes, with the amplitude of the lighter envelope decreasing with a commensurate increase in the heavier envelope over time (*Weis et al., 2006*). (ii) The fraction of the heavier envelope over time for prestin in Cl⁻ (black) and HEPES (three biological replicates are shown).

As just noted, the presenting data for TM1 point to conformational heterogeneity with two populations having distinct HDX behavior. We believe that the fast population has TM1 unfolded despite it having a PF of ~100. We attribute this residual protection to detergent molecules hindering solvent access to the backbone and hence slowing exchange. This is the same explanation we provided for the heightened protection observed for the N-terminal TM3 (Region$_{137-140}$) in apo prestin (*Figure 3A*). Generally, intrinsic HDX rates for unfolded regions of a soluble protein (i.e., $k_{chem}$) may not always serve as an appropriate reference rate for membrane-associated regions in the presence of detergents or lipids.

## Appendix 2

### Combining HDX-MS and cryo-EM in structural biology

HDX-MS can provide information on dynamics and thermodynamics that generally is unavailable with cryo-EM alone. Accordingly, the synergetic use of HDX-MS and cryo-EM can validate each other and provide new insights (*Engen and Komives, 2020*). We obtained a peptide coverage of 83% and 81% for prestin and SLC26A9, respectively, with a total of 266 and 338 peptides, allowing us to interrogate the protein-wide dynamics (*Appendix 2—figure 1*; *Table 1*). The main difference for the two proteins' sequence coverage is that only prestin has coverage at TM6 while only SLC26A9 has coverage at TM12. The different cleavage preferences at these two helices likely result from different flexiblity and/or exposure, which can be related to the different functions of the two proteins.

In 360 mM Cl$^-$, the majority of the TMDs for prestin and SLC26A9 had similar stability, with peptide-level PFs ranging from $10^3$ to $10^{6+}$; $\Delta G$ = 4.2 to 8.4+ kcal/mol (*Figure 2—figure supplement 1*, *Appendix 2—figure 1B*). Both proteins had highly stable regions with negligible exchange after 27 hr, our longest labeling time. We obtained near-residue level resolution at regions unresolved in the cryo-EM structures, including the intervening sequence of the STAS domain (sulfate transporter and anti-sigma factor antagonist) and the C-termini (*Bavi et al., 2021*; *Ge et al., 2021*; *Butan et al., 2022*; *Walter et al., 2019*; *Chi et al., 2020*). Regions$_{583–613, 734–764}$ for prestin and Regions$_{570–653, 741–770}$ for SLC26A9 had a PF of unity under all conditions examined (*Appendix 2—figure 2*), indicating these regions are unfolded and independent of anion binding. This finding supports the proposal that the disordered regions of prestin may play a role in its interactions with other proteins for reasons of regulation rather than electromotility (*Keller et al., 2014*; *Takahashi et al., 2019*).

HDX is a solution-based method that probes the hydrogen bond network, and hence, any discrepancies with cryo-EM structures could reflect structural perturbations resulting from the sub-millisecond freezing process (*Engstrom et al., 2021*). According to differences in cryo-EM and HDX, 2–3 residues form additional hydrogen bonds at the termini of two helices upon freezing. These sites included residues 565–566 and 720–722 for prestin and residues 225–226 and 738–740 for SLC26A9. Helical propagation can occur within 10 nsec (*Lin and Gai, 2017*) while the cooling time for cryo-EM can be as slow as 200 µs (*Engstrom et al., 2021*), which provides ample time for helix extension as the helices equilibrate to the lower temperature (*Bock and Grubmüller, 2022*). Although we anticipate that such small-scale folding events are common during the freezing process, overall, the HDX data do not provide evidence for significant changes in hydrogen bonding patterns for either prestin or SLC26A9. We anticipate that both the barriers for larger-scale folding events and solvent viscosity increase significantly as the temperature drops, effectively trapping the protein in its pre-frozen conformation. Nevertheless, we note that motions that do not result in changes in the hydrogen bond network, such as rigid-body motions of helices, would not be identified by HDX.

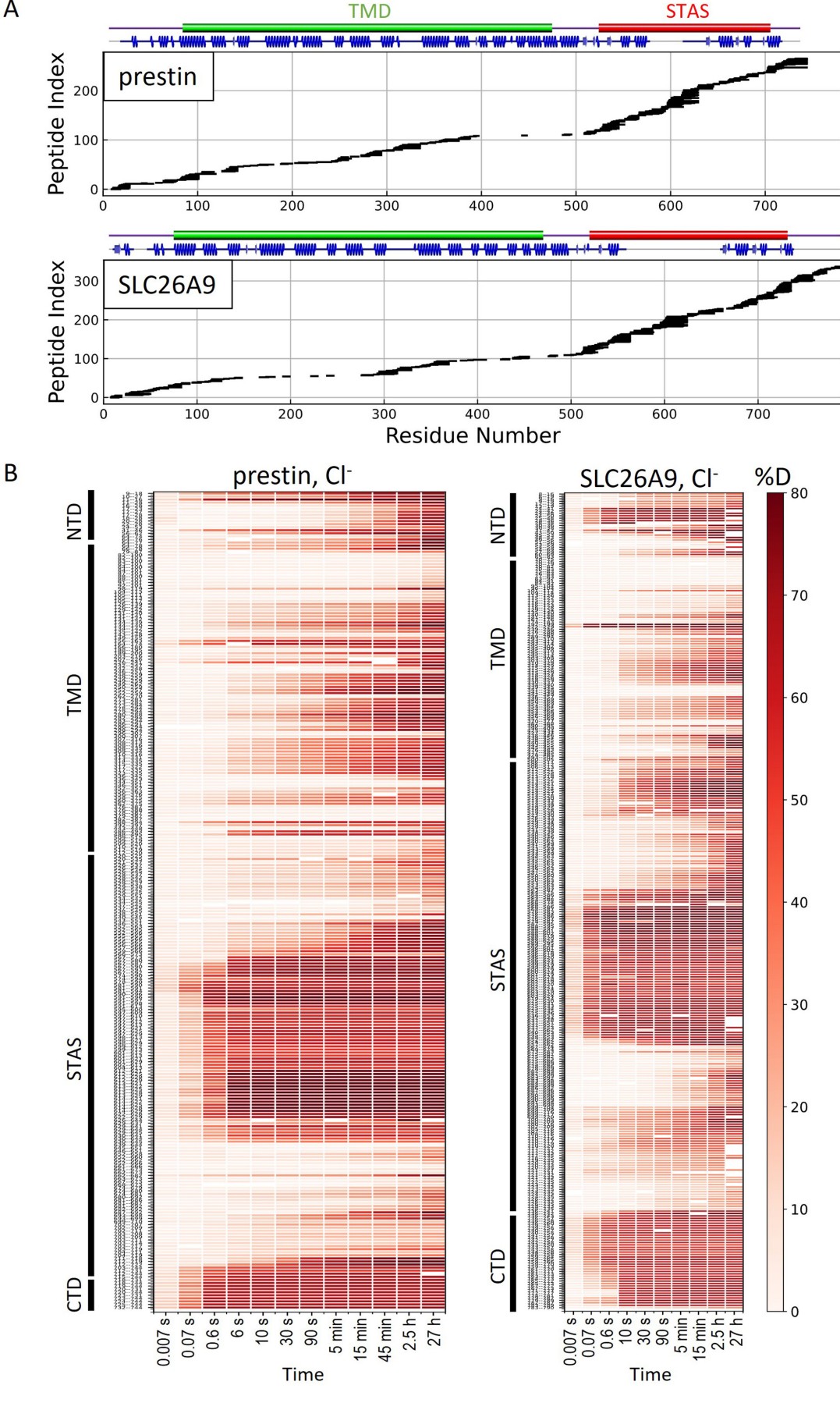

**Appendix 2—figure 1.** HDX-MS sequence coverage and measurements for prestin and SLC26A9 in Cl⁻. (**A**) Peptide sequence coverage for prestin and SLC26A9 suitable for HDX-MS analysis. On the top indicates the sequence boundary for domains and secondary structures. (**B**) Heatmaps showing deuteration levels of all the peptides at each labeling time for prestin and SLC26A9 measured in Cl⁻. Peptide sequences are displayed on the *y*-axis and legible through high-resolution images.

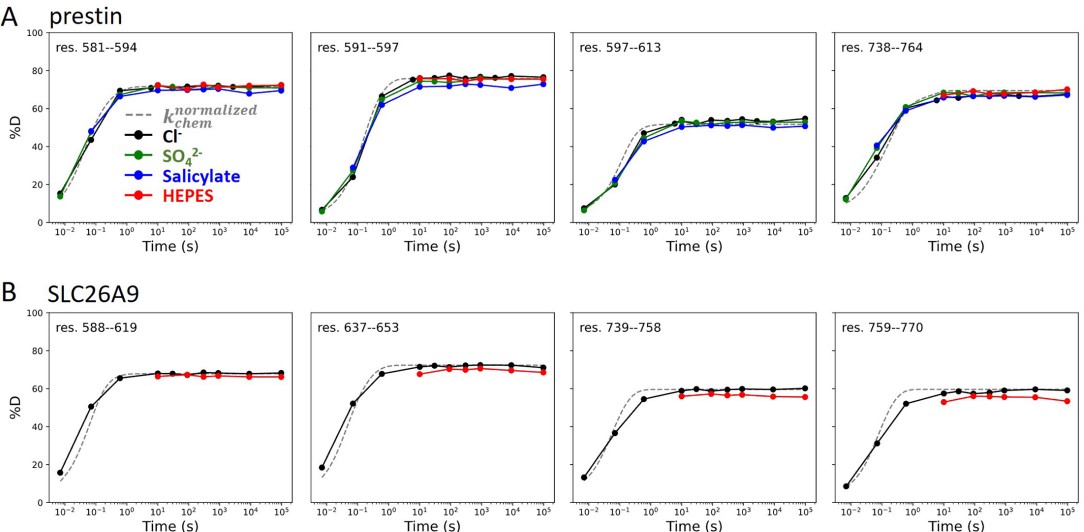

**Appendix 2—figure 2.** Regions unresolved in cryo-EM structures are unfolded in all conditions examined. Deuterium uptake plots for example peptides covering regions unresolved in cryo-EM structures for (**A**) prestin and (**B**) SLC26A9: black, Cl⁻; green: $SO_4^{2-}$; blue: salicylate; red: HEPES. Gray dashed curves represent deuterium uptake with $k_{chem}$, normalized with the in- and back-exchange levels. Only one replicate for Cl⁻ and HEPES are shown for clarity.

# Appendix 3

## Structure of prestin in HEPES and low Cl⁻ levels

Using single-particle cryo-EM, we set out to determine the structure of prestin in the HEPES-based buffer with the goal of visualizing a putative apo state. Prestin was initially screened in 190 mM HEPES in a nominal absence of $Cl^-$. This condition, however, led to widespread particle aggregation under cryogenic conditions. We reasoned that this aggregation may be linked to the already destabilized prestin without a bound $Cl^-$, as evidenced by HDX-MS data, as well as the low ionic strength of our buffer. Indeed, 1 mM $Cl^-$ sharply reduced particle aggregates, allowing us to solve the structure of prestin solubilized in GDN at a nominal resolution of 3.4 Å from particles, which corresponds to about 10% of the total particles in the sample (**Appendix 3—table 1**, **Appendix 3—figure 2**). Surprisingly, under these conditions prestin adopted a 'compact' conformation, virtually identical to the previously reported $Cl^-$-bound 'Up' state (**Appendix 3—figure 1**). Moreover, the anion-binding site is structurally indistinguishable from previous $Cl^-$-bound structures (**Appendix 3—figure 1**). Furthermore, when focusing on the anion-binding pocket in our cryo-EM map, we see clear evidence for an additional density, indicating that the pocket is occupied by a substrate (**Appendix 3—figure 1**). However, we were unable to model a HEPES anion into the binding pocket without substantial steric clashes. We therefore suggest that the resolved density represents instead a $Cl^-$ anion, given that a small population of $Cl^-$-bound prestin will be present from a weak $Cl^-$ affinity (e.g., $EC_{50} = 6$ mM[11] implies 17% bound). Although we cannot confirm this notion with absolute certainty, it is clear that our cryo-EM structure does not represent a true apo state of prestin and is consistent with the notion that unbound, apo prestin is conformationally unstable.

**Appendix 3—table 1.** Cryo-EM data collection, refinement, and validation statistics.

Data collection and processing

| | |
|---|---|
| Magnification | 81,000 |
| Voltage (kV) | 300 |
| Electron exposure (e⁻/Å²) | 60 |
| Defocus range (μm) | 0.7–2.1 |
| Pixel size (Å) | 1.068 |
| Symmetry imposed | C2 |
| Initial particle images (no.) | 328,033 |
| Final particle images (no.) | 170,313 |
| Map resolution (Å)<br>FSC threshold 0.143 | 3.4 |
| Map resolution range (Å) | 2.9–6.0 |
| Refinement | |
| Model resolution (Å)<br>0.5 FSC threshold | 3.7 |
| Model resolution range (Å) | n/a |
| Map sharpening $b$-factor (Å²) | 140.0 |
| Model composition<br>Non-hydrogen atoms<br>Protein residues<br>Ligands | 10,310<br>1336<br>2 |
| $B$ factors (Å²)<br>Protein<br>Ligands | 33.6<br>30 |
| R.m.s. deviations<br>Bond lengths (Å)<br>Bond angles (°) | 0.006<br>1.022 |

*Appendix 3—table 1 Continued on next page*

*Appendix 3—table 1 Continued*

Data collection and processing

| | |
|---|---|
| **Validation** | |
| MolProbity score | 1.98 |
| Clashscore | 14.14 |
| Poor rotamers (%) | 0.71 |
| | |
| **Ramachandran plot** | |
| Favored (%) | 95.78 |
| Allowed (%) | 4.22 |
| Disallowed (%) | 0 |

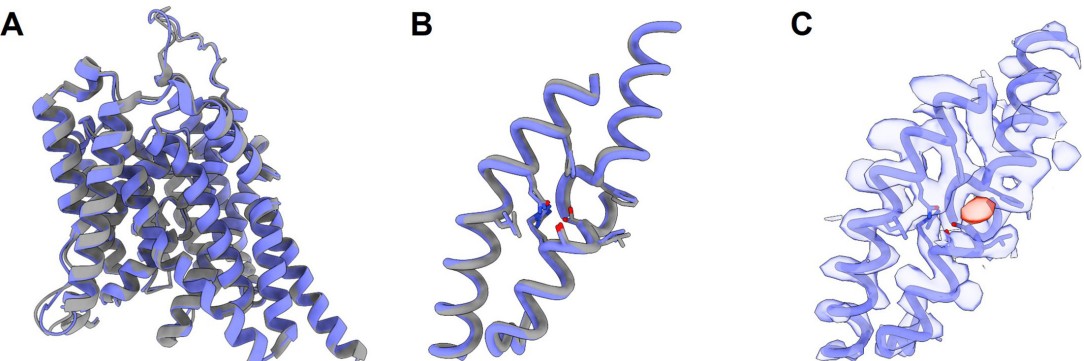

**Appendix 3—figure 1.** The cryo-EM structure for prestin in a HEPES-based buffer containing 1 mM Cl⁻ (blue; PDB 8UC1) highly resembles the structure in the reported Cl⁻-bound state. (**A**) Overlay of prestin's transmembrane domain (TMD) with that solved in a high-chloride buffer (gray; PDB 7S8X). (**B**) Overlay of TM1, TM3, and TM10, with key residues that make up the anion-binding site. (**C**) Cryo-EM density forming the anion-binding site (blue). Additional density (red) that is incompatible with the placing of a HEPES molecule was resolved at the anion-binding site.

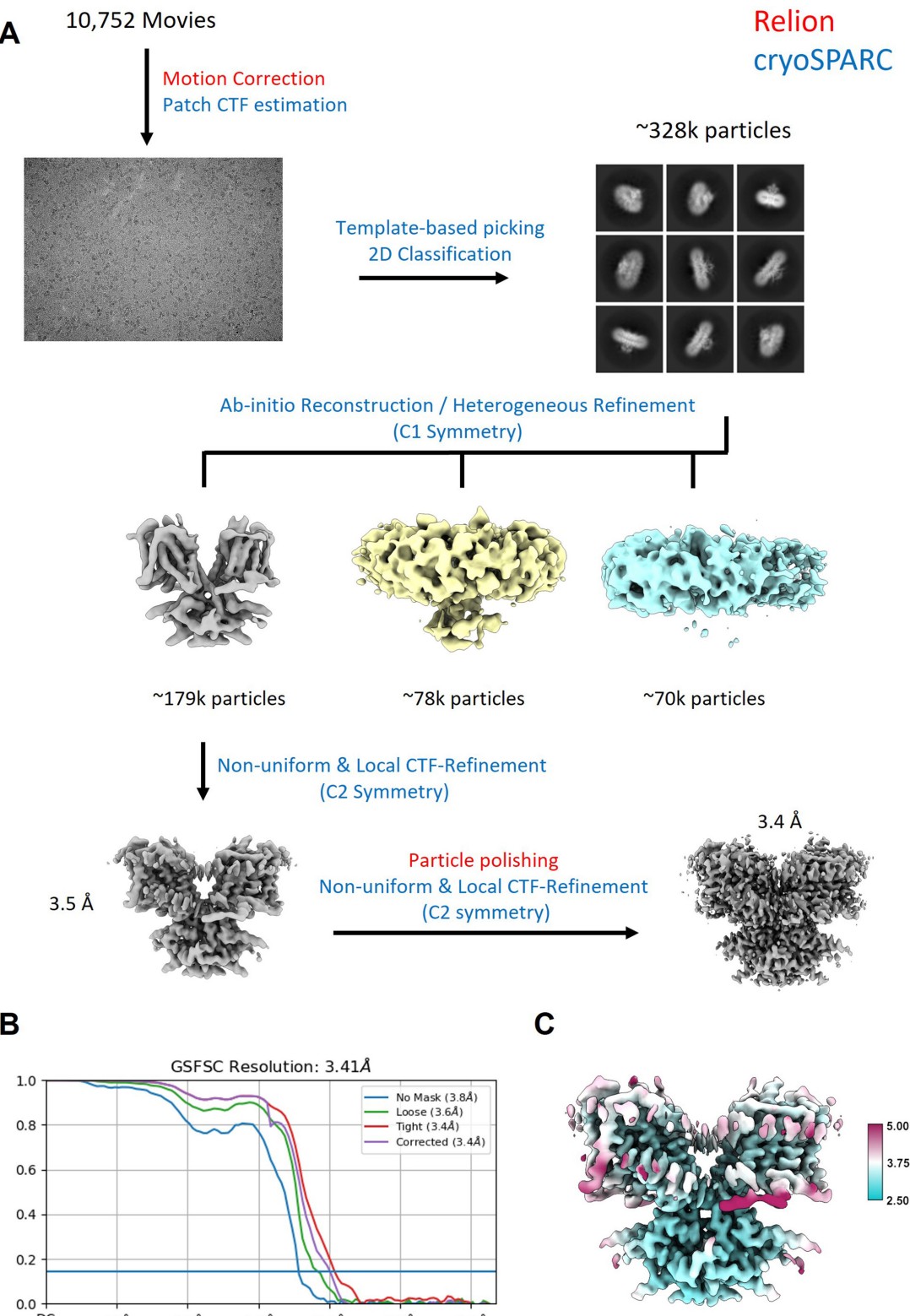

**Appendix 3—figure 2.** Workflow for the processing of the cryo-EM data. (**A**) Steps indicated in red font were performed in Relion, steps indicated in blue were performed in cryoSPARC. (**B**) FSC curve showing that the final reconstruction reached a nominal resolution of 3.4 Å (at FSC = 0.143). (**C**) Local resolution estimation of the final reconstruction.

