## [Editor Report · eLife assessment]

This study presents **important** findings regarding the local dynamics at the anion binding site in the SLC26 transporter prestin that is responsible for electromotility in outer hair cells. The authors reveal critical differences to homologous proteins and thereby provide insight into prestin's unique function. The evidence is generally **convincing**, although the interpretations concerning the mechanistic basis for voltage sensitivity would benefit from orthogonal evidence.

---

## [Referee Report · Reviewer #1 (Public Review)]

The manuscript by Lin, Sosnick et al investigates the functional conformational dynamics of two members of the SLC26 family of anion transporters (Prestin and SLC26A9). A key aspect of the work is that the authors use HDX-MS to convincingly identify that the folding of the unstable anion binding site is related to the fast electromechanical changes that are important for the function of Prestin. In good apparent agreement, such folding-related changes upon anion binding are absent in the related non-piezoelectric SLC26A9 that does not exhibit similar electro-motile transport. Overall, I find the work very interesting and generally well carried out - and it should be of considerable interest to researchers studying transmembrane transporters or just membrane proteins in general.

---

## [Referee Report · Reviewer #2 (Public Review)]

In this manuscript, Xiaoxuan Lin and colleagues provide new insights into the dynamics of prestin using H/D exchange coupled with mass spectrometry. The authors aim to reveal how local changes in folding upon anion binding sustain the unique electro-transduction capabilities of prestin.

Prestin is an unusual member of the SLC26 family, that changes its cross-sectional area in the membrane upon binding of a chloride ion. In contrast to SLC26 homologs, prestin is not an anion transporter per se but requires an anion to sense voltage. Binding of Cl- at a conserved binding site located between the end of TM3 and TM10 drives the displacement of a conserved arginine (R399), that causes major conformational changes, transmitting the voltage sensing into a mechanical force exerted on the membrane.

Cryo-EM structures are available for the protein bound to various anions, including Cl-, but these structures do not explain how a conserved couple of positive (R399) and negative (the Cl- anion) charge pair transforms voltage sensitivity into mechanical changes in the membrane. To address this challenge, the authors explore local dynamics of the anion binding site and compare it with that of a "real" anion transporter SLC26A9. The authors make a convincing case that the differences in local dynamics they measure are the molecular basis for voltage sensing and its translation into electromotility.

Practically the authors make a thorough HDX-MS investigation of prestin in the presence of different anions Cl-, SO4-, salicylate as well as in the apo form, and provide insight mostly on local dynamics of the anion binding site. The experiments are well-designed and conducted and their quality and reproducibility allows for quantitative interpretation by deriving ΔΔG values of changes in dynamics at specific sites. Furthermore, the authors show by comparing the apo condition with Cl- bound condition that the absence of Cl- causes fraying of the TM3 and TM10 helices. They deduce that Cl- binding allows for directional helix structuration, leading to local structural changes that cause a rearrangement of the charge configuration at the anion binding site that lays the molecular basis for voltage sensitivity. They demonstrate based on a detailed analysis of their HDX data that such helix fraying is a specific feature of the binding site and differs from the cooperative unfolding happening elsewhere on the prestin.

However, the main question that the authors are addressing is how voltage sensitivity translates at the molecular level in the requirement for a negative-positive charge pair. The interpretation that the binding site instability observed only for prestin is a feature required for this voltage dependent is a bit speculative. Could other lines of evidence support the claim that the charge ion gap is reduced upon Cl- binding and that this leads to cross-section area expansion? An obvious option that comes to mind is MD simulations There are differences in time-scale between HDX and simulations, but the propensity for H-bond destabilization can be quantified even at short timescales. It might be that such data is already available out there but it should be explicit in the discussion. The discussion section itself is a bit narrow in scope at the moment. Discussing the data in the context of the available structures would help the non-specialist reader.

---

## [Referee Report · Reviewer #3 (Public Review)]

Synopsis:

The lack of visualizing the dynamic nature of biomolecules is a major weakness of crystallography or electron microscopy to study structure-function relationship of proteins. Such a challenge can be exemplified by the case of prestin, which shares high structural similarity to SLC26A9 anion transporter but is not an ion transporter. In this study, Lin et al aimed to use hydrogen-deuterium exchange and mass spectrometry (HDX-MS) to investigate the mobility of prestin and its response to anions. The authors exploited the nature of anion-dependent folding of this type of transporter to systematically analyze the mobility of transmembrane helices of both transporters by HDX. The authors found that the anion-binding helices engage in the stabilization of the anion-binding site. When stripped from Cl-, the site exposes to the transporter's extracellular side. More importantly, the authors narrowed down TM3 and TM10 with experimental data supporting the notion of R399's unique role in prestin's function. The results thus provide a working model of how the charged residue works in conjunction with the cooperativity of helix unfolding at the anion-binding site to drive the electromotive force of prestin.

Strengths:

The use of HDX-MS to probe the dynamic nature of prestin is a major strength of this study, which provides experimental evidence revealing the global and local differences in the folding events between prestin and SLC26A9. The mass experimental data led to the identification of TM3 and TM10 as the primary contributors to the folding changes, as well as a calculation of ΔΔG of ~2.4 kcal/mol, within the thermodynamic range of the dipole between the two helices. The latter also suggests the role of R399 as previously speculated in cryo-EM structures.

This study went further to dissect the cooperativity during the folding and unfolding events on TM3, in which the authors observed a helix fraying at the anion-binding site and cooperative unfolding at the distal lipid-facing helices. This provides strong evidence of why prestin can undergo fast electromechanical rearrangement.

Weakness:

The authors tried to investigate the allostery by probing the intermediate folding/unfolding states by using sulfate or salicylate in the absence of chloride. Sulfate-bound proteins appear in an apo state earlier than normal chloride binding, and salicylate treatment led to a stable TMD state with slower HDX. It is unclear from the data (Fig 4) how the allostery works without titrating chloride ions into the reaction. The sulfate or salicylate experiments seem to show two extreme folding events outside the normal chloride conditions.

TM3 and TM10 contribute to the anion-binding site together, and the authors beautifully showed the cooperativity of TM3. Does TM10 show the same cooperativity in prestin and SLC26A9? In addition, it is unclear whether the folding model at the anion-binding helices (Fig. 5B) remains the same when expressing prestin on live cells, such as thermodynamic data derived from electrophysiology studies.

The authors observed increased stability upon chloride binding at the subunit interface in the cytosol for both prestin and SLC26A9 (Fig 1). How does this similarity in the cytosolic region contribute to the differential mechanisms as seen in the TMD in both transporters? It is unclear in this version of the manuscript.

---

## [Author Response]

The following is the authors’ response to the original reviews.

We thank the reviewers for their positive remarks. We have addressed the reviewers’ recommendations in the point-by-point response below to improve our revised manuscript.

**Reviewer #1 (Recommendations For The Authors):**
1. The authors carry out their HDX-MS work on Prestin (and SLC26A9) solubilized in glycol-diosgenin. The authors should carefully rationalize their choice of detergent and discuss how their key findings are also pertinent to the native state of Prestin when residing in an actual phospholipid bilayer. More native membrane mimetic models are available, for instance, nano-discs etc. While I am not insisting that the authors have to repeat their measurements in a more native membrane system, it would be a very nice control experiment, and in any case, a detailed discussion of the limitations of the approach taken and possible caveats should be included - possibly with additional references to other studies.

Response: We have added a paragraph rationalizing the choice of detergent in lines 174-176. We have also added requested HDX data comparing prestin reconstituted in nanodisc to prestin solubilized in micelle (Fig 5). The HDX for prestin under these two membrane mimetics were indistinguishable, including the anion-binding site, suggesting that our major findings are likely pertinent to prestin residing in a lipid bilayer. The only major HDX difference we observed was that a lipid-facing helix TM6 is more dynamic for prestin in nanodisc compared to in micelles. In our previous structural studies, we identified TM6 as the “eletromotile elbow” that is important for prestin’s mechanical expansion (Bavi et al., Nature, 2021). We are currently conducting a more thorough investigation to understand the role of TM6 in prestin’s electromotility.

1. As far as I understand, the HEPES state represents the apo-state and thus assumes that HEPES does not bind to Prestin - the authors should support this assumption or include a discussion of the possible effect of HEPES on Prestin. Also, the HEPES state has fewer time-points - this should also be discussed.

Response: We have included a discussion of the possible effects of HEPES in lines 331-345. In fact, in an attempt to support our assumption that HEPES does not bind to prestin, we set out to determine the structure of prestin in the HEPES-based buffer using single particle cryo-EM. However, we did not find evidence that HEPES binds to prestin. Details are discussed in lines 331-345 and Supporting Information Text 3.

We employed a denser sampling of HDX labeling times for prestin in Cl- because it is critical for fitting and ∆G calculation. The earlier time points are used mainly to evaluate the dynamics of the less stable cytosolic domain. Since the cytosolic domain does not directly participate in prestin’s voltage-sensing mechanism and electromotility, we only measured the HEPES states with longer time points which mainly probe the dynamics of the transmembrane domain.

1. Overall, the HDX-MS data provided and the statistical analysis done is in my view sufficiently detailed and well done - the authors are advised to make reference to and include a HDX Summary table and HDX Data Table according to the HDX-MS community-guidelines (Masson et al. Nature Methods 2019).

Response: An HDX summary table was provided in Table S1 and referred in lines 81 and 388. We have included a reference to Masson et al., Nature Methods, 2019, in line 389.

1. Figure 5 - I like the detailed analysis of the helix folding - but in my experience, one can provide a great fit of many HDX curves to a 4 -term exponential function - I think the authors would need more time-points to provide a more convincing case. But it does provide a compelling theory - even if the data strictly does not prove it. The authors should discuss this in more detail - including limitations etc.

Response: We presented a statistical analysis describing the accuracy of the fitting in Fig 6A. We acknowledge that the values of the exponentials may not be precisely determined, but the fundamental result is robust – TM3 exchanges through fraying from the N-terminal end of the helix while TM6 exchanges much more cooperatively. Collecting additional time points may reduce the error on the rates but would not contribute to additional mechanistic insights.

**Reviewer #2 (Recommendations For The Authors):**
1. I suggest toning down more speculative/ hypothetical aspects. Specifically, I believe that the following sentence should not be in the abstract in its present form: "This event shortens the TM3-TM10 electrostatic gap, thereby connecting the two helices such that TM3-anion-TM10 is pushed upwards by forces from the electric field, resulting in reduced cross-sectional area."

Response: The sentence has been rephrased.

1. The "nuance" between helix fraying and helix unfolding is an important aspect of the author's hypothesis but this should be explained better. In that regard, have the authors performed HDX-MS analysis of the mutant P136T? That would nicely support their claim regarding the importance of helix fraying as being foundational to allow electromotility.

Response: More explanation for helix fraying and unfolding has been provided in the main text.We have not performed HDX-MS analysis of the mutant P136T. However, we performed molecular dynamics simulations using Upside, and consistently, showed that a P136T mutation in prestin results in a highly stabilized TM3 (Fig. S4B).

1. Why do measurements at two pDs? Did the authors observe any differences?

Response: The purpose of two pDs is to increase the effective dynamic range of the HDX measurement by two orders of magnitude because the intrinsic exchange rate scales with pD & Temp. This allows us to determine the stability of both the highly and minimally stable regions within the protein. We have rephrased lines 83-87 to better rationalize this choice of pDs. With the time points performed in this study, we did not observe noticeable differences for HDX performed under the two pDs when corrected for the changes in the intrinsic rates (Fig. S7A).

1. I can't help but wonder what is the interest in doing HDX-MS measurements after 27h of incubation. Membrane proteins are known for their instability once purified and a few odd HDX profiles at that specific timepoint (especially in the 80-100 residues area) make one question whether local unfolding preceding aggregation could happen. This actually weakens the author's claims about cooperative unfolding and localized and directional helix fraying. Could they provide some evidence (CD, thermostability measurements such as trp fluorescence quenching, or SEC analysis) that the prestin is still folded after 27h in GDN.

Response: We appreciate reviewer’s comments on membrane proteins can be unstable once purified. In our system, we did not observe evidence of unfolding or aggregation caused by long-term incubation after purification. This is mostly supported by the fact that our HDX reactions were initiated and injected to MS in random order, yet are still highly reproducible among biological and technical replicates. A specific example included HDX on freshly purified SLC26A9 gave the same deuteration levels as SLC26A9 purified in GDN after 4 days. For prestin, although we don’t have direct comparison between fresh samples and old samples (24-27h post-purification) due to the lack of samples, 30s HDX in SO42- performed 24h post-purification gave a %D that fell between 10s and 90s of labeling done on fresh sample. Additionally, HDX on prestin in Cl- performed on freshly purified sample gave the sample %D as prestin in the presence of 1M urea labeled after 24~48h of purification, suggesting that prestin is relatively resistant to aggregation at least within 48h after purification even in the presence of 1 M urea (data not shown).

Furthermore, the HDX for prestin in nanodisc are essentially identical to prestin in micelles except for a functionally important helix (TM6), suggesting minimal aggregation or misfolding.

We think the “a few odd HDX profiles” at 27h time points for residues 80-100 are caused by two reasons. Firstly, TM1 unfolds cooperatively and its stability in HEPES falls within the detection range when long labeling time points were employed (within one log unit of 27h). Secondly, we observed two non-interconverting and structurally distinct populations for TM1 (Supporting Information Text 1 & Fig. S8), and in long labeling times, the two isotope distributions merge and sometimes can skew the %D calculations. Nevertheless, the HDX differences we observed comparing across conditions are clear and such %D calculation skewing, if present, should be minimal and does not change our main conclusions.